# FreeSplat: Generalizable 3D Gaussian Splatting Towards Free-View Synthesis of Indoor Scenes

**Yunsong Wang**   **Tianxin Huang**   **Hanlin Chen**   **Gim Hee Lee**
School of Computing, National University of Singapore
yunsong@comp.nus.edu.sg   gimhee.lee@nus.edu.sg
https://github.com/wangys16/FreeSplat

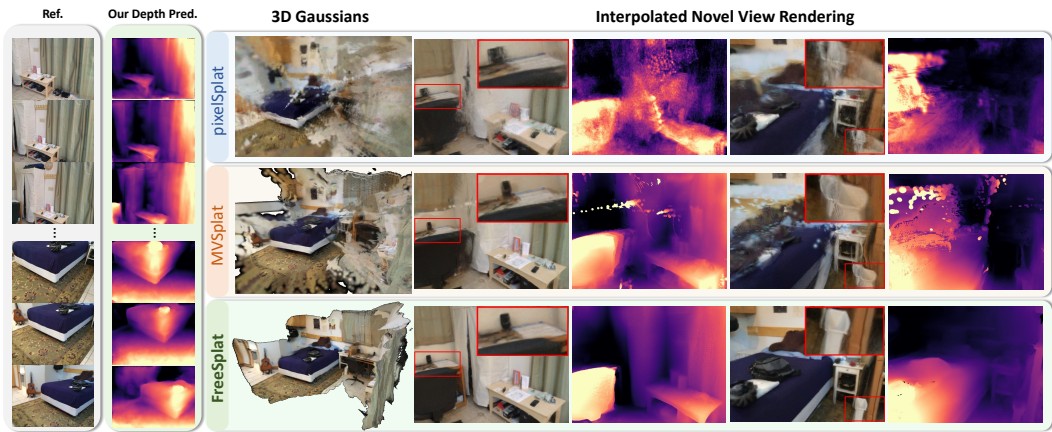

Figure 1: **Comparison between FreeSplat and previous methods.** pixelSplat [1] and MVSplat [2] fail to reconstruct geometrically consistent global 3D Gaussians, while our FreeSplat is proposed to accurately localize 3D Gaussians from long sequence input and support free view synthesis.

## Abstract

Empowering 3D Gaussian Splatting with generalization ability is appealing. However, existing generalizable 3D Gaussian Splatting methods are largely confined to narrow-range interpolation between stereo images due to their heavy backbones, thus lacking the ability to accurately localize 3D Gaussian and support free-view synthesis across wide view range. In this paper, we present a novel framework FreeSplat that is capable of reconstructing geometrically consistent 3D scenes from long sequence input towards free-view synthesis. Specifically, we firstly introduce Low-cost Cross-View Aggregation achieved by constructing adaptive cost volumes among nearby views and aggregating features using a multi-scale structure. Subsequently, we present the Pixel-wise Triplet Fusion to eliminate redundancy of 3D Gaussians in overlapping view regions and to aggregate features observed across multiple views. Additionally, we propose a simple but effective free-view training strategy that ensures robust view synthesis across broader view range regardless of the number of views. Our empirical results demonstrate state-of-the-art novel view synthesis peformances in both novel view rendered color maps quality and depth maps accuracy across different numbers of input views. We also show that FreeSplat performs inference more efficiently and can effectively reduce redundant Gaussians, offering the possibility of feed-forward large scene reconstruction without depth priors.

38th Conference on Neural Information Processing Systems (NeurIPS 2024).

# 1   Introduction

Recent advancements has emerged [3, 4, 5, 6] in reconstructing 3D scenes from multiple viewpoints. Based on ray-marching-based volume rendering, Neural Radiance Fields [3, 7, 8, 9] is capable of learning the implicit 3D geometry and radiance fields without depth information. Nonetheless, computational cost remains to be the inherent bottleneck in ray-marching-based volume rendering, preventing it from real-time rendering. 3D Gaussian Splatting [10, 11, 12, 13] has recently been proposed as an efficient representation for photorealistic reconstruction of 3D scenes from multi-views. The explicit representation of 3D Gaussians are optimized to be densified in the textured regions, and the rasterization-based volume rendering avoids the costly ray marching scheme. Consequently, 3D Gaussian Splatting has achieved real-time rendering of high-quality images from novel views. Nonetheless, vanilla 3D Gaussian Splatting lacks generalizability and requires per-scene optimization.

Several attempts [1, 2, 14, 15, 16, 17] have been made to give 3D Gaussian Splatting generalization ability. Despite showing promising performance, these methods are limited to narrow-range scene-level view interpolation [1, 2, 15] and object-centric synthesis [14, 16]. The primary reason for the limitation is that these existing methods depend on dense view matching across multi-view images with transformers to predict Gaussian primitives, which consequently becomes computationally intractable with longer sequences and thus restricting the supervision of these methods to narrow-range interpolated views. As we show in Figure 4, supervision by narrow-range interpolated views often result in poorly localized 3D Gaussians that can become floaters when rendered from extrapolated views. Additionally, the problem is further aggrevated by existing methods typically merging multi-view 3D Gaussians through simple concatenation and thus inevitably lead to noticeable redundancy in overlapping areas (*cf.* Table 2). In view of the above-mentioned problems, it is therefore imperative to design a method that is capable of long sequence reconstruction of global 3D Gaussians, which has the significant potential of supporting real-time rendering from arbitrary poses.

In this paper, we propose FreeSplat tailored for indoor long sequence free view synthesis. Unlike existing methods limited to view interpolation in narrow ranges, our method can effectively reconstruct explicit global 3DGS for novel view synthesis across wide view ranges. Our pipline consists of Low-cost Cross-View Aggregation and Pixel-wise Triplet Fusion (PTF). In Low-cost Cross-View Aggregation, we introduce efficient CNN-based backbones and adaptive cost volumes formulation among nearby views for low-cost feature extraction and matching, then we leverage a Multi-Scale Feature Aggregation structure to broaden the receptive field of cost volume and predict Depths and Gaussian Triplets. Subsequently, we present Pixel-wise Alignment with progressive Gaussian fusion in PTF to adaptively fuse local Gaussian Triplets from multi-views and avoid Gaussian redundancy in the overlapping regions. Moreover, due to our efficient feature extraction and matching, we propose a Free-View Training (FVT) strategy to disentangle generalizable 3DGS with specific number of views and train the model on long sequences.

The **contributions** of our paper are summarized as follows:

1. We present Low-cost Cross-View Aggregation to predict initial Gaussian triplets, where the low computational cost makes it possible for feature matching between more nearby views and training on long sequence reconstruction;

2. We propose Pixel-wise Triplet Fusion to fuse Gaussian triplets, which can effectively reduce the Gaussian redundancy in the overlapping regions and aggregate multi-view 3D Gaussian latent features;

3. To the best of our knowledge, we are the first to explore generalizable 3DGS for long sequence reconstruction. Extensive experiments on indoor dataset ScanNet [18] and Replica [19] demonstrate our superiority on both image rendering quality and novel view depth rendering accuracy when given different lengths of input views.

# 2   Related Work

**Novel View Synthesis.** Traditional attempts in novel view synthesis mainly employed voxel grids [20, 21] or multiplane images [22]. Recently, Neural Radiance Fields (NeRF) [3, 5, 23, 24, 25] have drawn growing interest using ray-marching-based volume rendering to backpropagate image color error to the implicit geometry and radiance fields, such that the 3D geometry can be implicitly learned

to satisfy the multi-view color consistency. Nonetheless, one inherent bottleneck of NeRFs-based method is the computation intensity of ray marching, which requires the costly volume sampling in the implicit fields for each pixel during rendering. To this end, recently 3DGS [10, 26, 27, 11] have attracted increasing attention due to its high efficiency and photorealistic rendering. Instead of relying on MLPs to represent the coordinate-based implicit fields, 3DGS learns an explicit field using a set of 3D Gaussians. They optimize the 3D Gaussians parameters and perform adaptive densify control to fit to the given set of images, such that the 3D Gaussians are encouraged to perform densification only in the textured regions and refrain from over-densification. During rendering, 3DGS performs tile-based rasterization to differentiably accumulate color images from the explicit 3D Gaussian primitives, which is significantly faster than the ray-marching-based volume rendering and achieves real-time rendering speed.

**Generalizable Novel View Synthesis.** Another drawback of the traditional NeRF-based and 3DGS-based methods is the requirement of per-scene optimization instead of direct feeding-forward. To this end, there have been a line of work [28, 8, 7, 29] focusing on learning effective priors to predict 3D geometry from given images in a feed forward fashion, where the common practice is to project ray-marching sampled points onto given source views to aggregate multi-view features, conditioning the prediction of the implicit fields on source views instead of point coordinates. Recently, there have also been attempts towards generalizable 3DGS [1, 17, 2, 15, 14]. pixelSplat [1] and GPS-Gaussian [17] propose to predict pixel-aligned 3D Gaussian parameters in feed forward fashion. MVSplat [2] replaces the epipolar line transformer of pixelSplat with a lightweight cost volume to perform more efficient image encoding. GGRt [15] concatenates pixelSplat predicted 3D Gaussians in a sequence of images and simultaneously perform pose optimization. latentSplat [14] encodes 3D Variational Gaussians and leverages a discriminator to help produce more indistinguable images. Nonetheless, existing methods do not reconstruct the global 3D Gaussians from arbitrary length of inputs, and are limited to view interpolation [1, 2, 17, 15] or object/human-centric scenes [17, 14]. In contrary, in this paper we focus on reconstructing large scenes from arbitrary length of inputs without depth priors, unleashing the potential of generalizable 3DGS for large scene explicit representation.

**Indoor Scene Reconstruction.** One line of efforts in feed-forward indoor scene reconstruction focuses on extracting 3D mesh using voxel volumes [30, 31, 32] and TSDF-fusion [33], while do not perform photorealistic novel view synthesis. On the other hand, the SLAM-based methods [34, 35, 36] require dense sequence of RGB-D input and per-scene tracking and mapping. Another paradigm of 3D reconstruction [37, 38, 39] learns implicit Signed Distance Fields from RGB input, while demanding intensive per-scene optimization. Another recent work SurfelNeRF [40] learns a feed-forward framework to map a sequence of images to 3D surfels which support photorealistic image rendering, while they rely on external depth estimator or ground truth depth maps. In contrary, we propose an end-to-end model without ground truth depth map input or supervision, enabling accurate 3D Gaussian localization using only photometric losses.

## 3 Preliminary

**Vanilla 3DGS.** 3D-GS [10] explicitly represents a 3D scene with a set of Gaussian primitives which are parameterized via a 3D covariance matrix $\Sigma$ and mean $\boldsymbol{\mu}$:

$$G(\mathbf{p}) = \exp(-\frac{1}{2}(\boldsymbol{p} - \boldsymbol{\mu})^\top \Sigma^{-1}(\boldsymbol{p} - \boldsymbol{\mu})), \tag{1}$$

where $\Sigma$ is decomposed into $\Sigma = \mathbf{R}\mathbf{S}\mathbf{S}^\top\mathbf{R}^\top$ using a scaling matrix $\mathbf{S}$ and a rotation matrix $\mathbf{R}$ to maintain positive semi-definiteness. During rendering, the 3D Gaussian is transformed into the image coordinates with world-to-camera transform matrix $\mathbf{W}$ and projected onto image plane with projection matrix $\mathbf{J}$, and the 2D covariance matrix $\Sigma'$ is computed as $\Sigma' = \mathbf{J}\mathbf{W}\Sigma\mathbf{W}^\top\mathbf{J}^\top$. We then obtain a 2D Gaussian $G^{2D}$ with the covariance $\Sigma'$ in 2D, and the color rendering is computed using point-based alpha-blending on each ray:

$$\mathbf{C}(\mathbf{x}) = \sum_{i \in N} \mathbf{c}_i \alpha_i G_i^{2D}(\mathbf{x}) \prod_{j=1}^{i-1}(1 - \alpha_j G_j^{2D}(\mathbf{x})), \tag{2}$$

where $N$ is the number of Gaussian primitives, $\alpha_i$ is a learnable opacity, and $\mathbf{c}_i$ is view-dependent color defined by spherical harmonics (SH) coefficients $\mathbf{s}$. The Gaussian parameters are optimized by a photometric loss to minimize the difference between renderings and image observations.

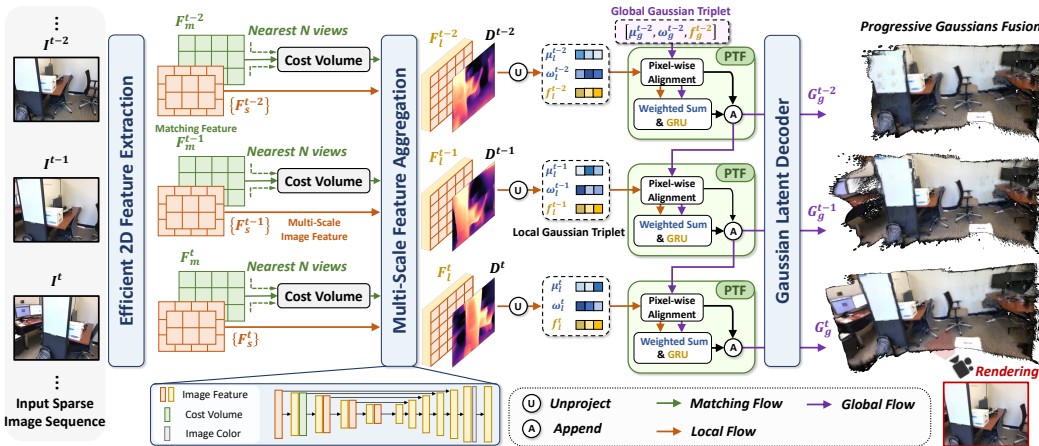

Figure 2: **Framework of FreeSplat.** Given input sparse sequence of images, we construct cost volumes between nearby views and predict depth maps and corresponding feature maps, followed by unprojection to Gaussian triplets with 3D positions. We then propose **Pixel-aligned Triplet Fusion (PTF)** module, where we progressively aggregate and update local/global Gaussian triplets based on pixel-wise alignment. The global Gaussian triplets can be later decoded into Gaussian parameters.

**Generalizable 3DGS.** Unlike vanilla 3DGS that optimizes per-scene Gaussian primitives, recent generalizable 3DGS [1, 17] predict pixel-aligned Gaussian primitives $\{\Sigma, \alpha, s\}$ and depths $d$, such that the pixel-aligned Gaussian primitives can be unprojected to 3D coordinates $\mu$. The Gaussian parameters are predicted by 2D encoders, which are optimized by the photometric loss through rendering from novel views. However, existing methods are still limited to view interpolation within narrow view range, which leads to inaccurately localized 3D Gaussians that fail to support large scene reconstruction and view extrapolation (*cf.* Figure 1, 4). To this end, we propose FreeSplat towards global 3D Gaussians reconstruction with accurate localization that supports free-view synthesis.

# 4 Our Methodology

## 4.1 Overview

The overview of our method is illustrated in Figure 2. Given a sparse sequence of RGB images, we build cost volumes adaptively between nearby views, and predict depth maps to unproject the 2D feature maps into 3D Gaussian triplets. We then propose the Pixel-aligned Triplet Fusion (PTF) module to progressively align the global with the local Gaussian triplets, such that we can fuse the redundant 3D Gaussians in the latent feature space and aggregate cross-view Gaussian features before decoding. Our method is capable of efficiently exchanging cross-view features through cost volumes, and progressively aggregating per-view 3D Gaussians with cross-view alignment and adaptive fusion.

## 4.2 Low-cost Cross-View Aggregation

**Efficient 2D Feature Extraction.** Given a sparse sequence of posed images $\{I^t\}_{t=1}^T$, we first feed them into a shared 2D backbone to extract multi-scale embeddings $F_e^t$ and matching feature $F_m^t$. Unlike [1, 2] which rely on patch-wise transformer-based backbones [41, 42] that can lead to quadratically expensive computations, we leverage pure CNN-based backbones [43, 44] for 2D feature extraction for efficient performance on higher resolution inputs.

**Adaptive Cost Volume Formulation.** To explicitly integrate camera pose information given arbitrary length of input images, we propose to adaptively build cost volumes between nearby views. For current view $I^t$ with pose $P^t$ and matching feature $F_m^t \in \mathbb{R}^{C_m \times \frac{H}{4} \times \frac{W}{4}}$, we adaptively select its $N$ nearby views $\{I^{t_n}\}_{n=1}^N$ with poses $\{P^{t_n}\}_{n=1}^N$ based on pose proximity, and construct cost volume via plane sweep stereo [45, 46]. Specifically, we define a set of $K$ virtual depth planes $\{d_k\}_{k=1}^K$ that are uniformly spaced within $[d_{near}, d_{far}]$, and warp the nearby view features to each depth plane $d_k$

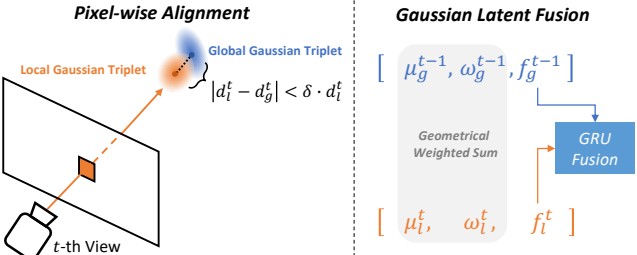

Figure 3: **Visual illustration of PTF.** The PTF incrementally projects current global Gaussians to input views and computes their pixel-wise distance with local Gaussians. Nearby local Gaussians are then fused using a lightweight Gate Recurrent Unit (GRU) network [49].

of current view:

$$\tilde{\boldsymbol{F}}_m^{t_n,k} = \mathrm{Trans}(\mathbf{P}^{t_n}, \mathbf{P}^t)\boldsymbol{F}_m^{t_n}, \tag{3}$$

where $\mathrm{Trans}(\mathbf{P}^{t_n}, \mathbf{P}^t)$ is the transformation matrix from view $t_n$ to $t$. The cost volume $\boldsymbol{F}_{\mathrm{cv}}^t \in \mathbb{R}^{K \times \frac{H}{4} \times \frac{W}{4}}$ is then defined as:

$$\boldsymbol{F}_{\mathrm{cv}}^t(k) = f_\theta \left( (\frac{1}{N} \sum_{n=1}^{N} \cos(\boldsymbol{F}_m^t, \tilde{\boldsymbol{F}}_m^{t_n,k})) \oplus (\frac{1}{N} \sum_{n=1}^{N} \tilde{\boldsymbol{F}}_m^{t_n,k}) \right), \tag{4}$$

where $\boldsymbol{F}_{\mathrm{cv}}^t[k]$ is the $k$-th dimension of $\boldsymbol{F}_{\mathrm{cv}}^t$, $\cos(\cdot)$ is the cosine similarity, $\oplus$ is feature-wise concatenation, and $f_\theta(\cdot)$ is a $1 \times 1$ CNN mapping to dimension of 1.

**Multi-Scale Feature Aggregation.** The embedding of the cost volume plays a significant part to accurately localize the 3D Gaussians (*cf.* Table 5). To this end, inspired by previous depth estimation methods [47, 33], we design an multi-scale encoder-decoder structure, such that to fuse multi-scale image features with the cost volume and propagate the cost volume information to broader receptive fields. Specifically, the multi-scale encoder takes in $\boldsymbol{F}_{\mathrm{cv}}^t$ and the output is concatenated with $\{\boldsymbol{F}_s^t\}$ before sending into a UNet++ [48]-like decoder to upsample to full resolution and predict a depth candidates map $\boldsymbol{D}_c^t \in \mathbb{R}^{K \times H \times W}$, and Gaussian triplet map $\boldsymbol{F}_l^t \in \mathbb{R}^{C \times H \times W}$. We then predict the depth map through soft-argmax to bound the depth prediction between near and far:

$$\boldsymbol{D}^t = \sum_{k=1}^{K} \mathrm{softmax}(\boldsymbol{D}_c^t)_k \cdot d_k. \tag{5}$$

Finally, the pixel-aligned Gaussian triplet map $\boldsymbol{F}_l^t$ is unprojected to 3D Gaussian triplet $\{\boldsymbol{\mu}_l^t, \boldsymbol{\omega}_l^t, \boldsymbol{f}_l^t\}$, where $\boldsymbol{\mu}_l^t \in \mathbb{R}^{3 \times HW}$ are the Gaussian centers, $\boldsymbol{\omega}_l^t \in \mathbb{R}^{1 \times HW}$ are weights between $(0, 1)$, and $\boldsymbol{f}_l^t \in \mathbb{R}^{(C-1) \times HW}$ are Gaussian triplet features.

### 4.3 Pixel-wise Triplet Fusion

One limitation of previous generalizable 3DGS methods is the redundancy of Gaussians. Since we need multi-view observations to predict accurately localized 3D Gaussians in indoor scenes, the pixel-aligned Gaussians become redundant in frequently observed regions. Furthermore, previous methods integrate multi-view Gaussians of the same region simply through their opacities, leading to suboptimal performance due to lack of post aggregation (*cf.* Table 5). Consequently, inspired by previous methods [31, 40], we propose the Pixel-wise Triplet Fusion (PTF) module which can significantly remove redundant Gaussians in the overlapping regions and explicitly aggregate multi-view observation features in the latent space. We align the per-view local Gaussians with global ones using Pixel-wise Alignment to select the redundant 3D Gaussian Triplets, and progressively fuse the local Gaussians into the global ones.

**Pixel-wise Alignment.** Given the Gaussian triplets $\{\boldsymbol{\mu}_l^t, \boldsymbol{f}_l^t\}_{t=1}^{T}$, we start from $t = 1$ where the global Gaussians latent is empty. In the $t$-th step, we first project the global Gaussian triplet centers $\boldsymbol{\mu}_g^{t-1} \in \mathbb{R}^{3 \times M}$ onto the $t$-th view:

$$\mathbf{p}_g^t := \{\mathbf{x}_g^t, \mathbf{y}_g^t, \mathbf{d}_g^t\} = \boldsymbol{P}^t \boldsymbol{\mu}_g^{t-1}, \tag{6}$$

where $[\mathbf{x}_g^t, \mathbf{y}_g^t, \mathbf{d}_g^t] \in \mathbb{R}^{3 \times M}$ are the projected 2D coordinates and corresponding depths. We then correspond the local Gaussian triplets with the pixel-wise nearest projections within a threshold. Specifically, for the $i$-th local Gaussian with 2D coordinate $[\mathbf{x}_l^t(i), \mathbf{y}_l^t(i)]$ and depth $\mathrm{d}_\mathrm{l}^\mathrm{t}(\mathrm{j})$, we first find its intra-pixel global projection set $\boldsymbol{\mathcal{S}}_i$:

$$\boldsymbol{\mathcal{S}}_i^t := \{j \mid [\mathbf{x}_g^t(j)] = \mathbf{x}_l^t(i), [\mathbf{y}_g^t(j)] = \mathbf{y}_l^t(i)\}, \tag{7}$$

where $[\,\cdot\,]$ is the rounding operator. Subsequently, we search for valid correspondence with minimum depth difference under a threshold:

$$m_i = \begin{cases} \underset{j \in \boldsymbol{\mathcal{S}}_i^t}{\arg\min} \, \mathbf{d}_g^t(j) & \text{if } \mid \mathbf{d}_l^t(j) - \underset{j \in \boldsymbol{\mathcal{S}}_i^t}{\min} \, \mathbf{d}_g^t(j) \mid < \delta \cdot \mathbf{d}_l^t(j) \\ \varnothing & \text{otherwise} \end{cases}, \tag{8}$$

where $\delta$ is a ratio threshold. We define the valid correspondence set as:

$$\boldsymbol{\mathcal{F}}^t := \{(i, m_i) \mid i = 1, ..., HW; \; m_i \neq \varnothing\}. \tag{9}$$

**Gaussian Triplet Fusion.** After the pixel-wise alignment, we remove the redundant 3D Gaussians through merging the validly aligned triplet pairs. Given a pair $(i, m_i) \in \mathcal{F}^\mathrm{t}$, we compute the weighted sum of their center coordinates and sum their weights to restrict the 3D Gaussian centers to lie between the triplet pair:

$$\boldsymbol{\mu}_g^t(m_i) = \frac{\boldsymbol{\omega}_l^t(i)\boldsymbol{\mu}_l^t(i) + \boldsymbol{\omega}_g^{t-1}(m_i)\boldsymbol{\mu}_g^{t-1}(m_i)}{\boldsymbol{\omega}_l^t(i) + \boldsymbol{\omega}_g^t(m_i)}, \quad \text{where } \boldsymbol{\omega}_g^t(m_i) = \boldsymbol{\omega}_l^t(i) + \boldsymbol{\omega}_g^{t-1}(m_i). \tag{10}$$

We then aggregate the aligned local and global Gaussian latent features through a lightweight GRU network:

$$\boldsymbol{f}_g^t(m_i) = \mathrm{GRU}(\boldsymbol{f}_l^t(i), \boldsymbol{f}_g^{t-1}(m_i)), \tag{11}$$

and then append with the other unaligned local Gaussian triplets.

**Gaussian primitives decoding.** After the Pixel-wise Triplet Fusion, we can decode the global Gaussian triplets into Gaussian primitives:

$$\boldsymbol{\Sigma}, \boldsymbol{\alpha}, \mathbf{s} = \mathrm{MLP}_d(\boldsymbol{f}_g^T) \tag{12}$$

and Gaussian centers $\boldsymbol{\mu} = \boldsymbol{\mu}_g^\top$. Our proposed fusion method can incrementally integrate the Gaussians with geometrical constraints and learnable GRU network for feature update. Consequently, our fusion method is capable of significantly removing redundant Gaussians and perform post feature aggregation across multiple views, and can be trained with the other framework components end-to-end with eligible computation overhead.

## 4.4 Training

**Loss Functions.** After predicting the 3D Gaussian primitives, we render from novel views following the rendering equations in Eq. (2). Similar to pixelSplat [1] and MVSplat [2], we train our framework using only photometric losses, *i.e.* a combination of MSE loss and LPIPS [50] loss, with weights of 1 and 0.05 following [1, 2].

**Free-View Training.** We propose a Free-View Training (FVT) strategy to add more geometrical constraints on the localization of 3D Gaussians, and to disentangle the performance of generalizable 3DGS with specific number of input views. To this end, we randomly sample $T$ number of context views (in experiments we set $T$ between 2 and 8), and supervise the image renderings in the broader view interpolations. The long sequence training is made feasible due to our efficient feature extraction and aggregation. We empirically find that FVT significantly contributes to depth estimation from novel views (*cf.* Table 3, 4).

## 5 Experiments

### 5.1 Experimental Settings

**Datasets.** We leverage the real-world indoor dataset ScanNet [18] for training. ScanNet is a large RGB-D dataset containing $1,513$ indoor scenes with camera poses, and we follow [51, 40] to use

Table 1: **Generalizable Novel View Interpolation results on ScanNet [18].** FreeSplat-*fv* is trained with our FVT strategy, and the other methods are all trained on specific number of views to form a complete comparison. Time(s) indicates the total time of encoding input images and rendering one image.

| Method | 2 views | | | | | 3 views | | | | |
| | PSNR↑ | SSIM↑ | LPIPS↓ | Time(s)↓ | #GS(k) | PSNR↑ | SSIM↑ | LPIPS↓ | Time(s)↓ | #GS(k) |
|---|---|---|---|---|---|---|---|---|---|---|
| NeuRay [9] | 25.65 | 0.840 | 0.264 | 3.103 | - | 25.47 | 0.843 | 0.264 | 4.278 | - |
| pixelSplat [1] | 26.03 | 0.784 | 0.265 | 0.289 | 1180 | 25.76 | 0.782 | 0.270 | 0.272 | 1769 |
| MVSplat [2] | 27.27 | 0.822 | 0.221 | 0.117 | 393 | 26.68 | 0.814 | 0.235 | 0.192 | 590 |
| **FreeSplat-*spec*** | 28.08 | 0.837 | 0.211 | 0.103 | 278 | 27.45 | 0.829 | 0.222 | 0.121 | 382 |
| **FreeSplat-*fv*** | 27.67 | 0.830 | 0.215 | 0.104 | 279 | 27.34 | 0.826 | 0.226 | 0.122 | 390 |

Table 2: **Long Sequence (10 views) Explicit Reconstruction results on ScanNet.** The results of pixelSplat, MVSplat and FreeSplat-*spec* are given using their 3-views version.

| Method | Time(s)↓ | #GS(k) | View Interpolation | | | View Extrapolation | | |
| | | | PSNR↑ | SSIM↑ | LPIPS↓ | PSNR↑ | SSIM↑ | LPIPS↓ |
|---|---|---|---|---|---|---|---|---|
| pixelSplat [1] | 0.948 | 5898 | 21.26 | 0.714 | 0.396 | 20.70 | 0.687 | 0.429 |
| MVSplat [2] | 1.178 | 1966 | 22.78 | 0.754 | 0.335 | 21.60 | 0.729 | 0.365 |
| **FreeSplat-*3views*** | 0.599 | 882 | 25.15 | 0.800 | 0.278 | 23.78 | 0.774 | 0.309 |
| **FreeSplat-*fv*** | 0.596 | 899 | 25.90 | 0.808 | 0.252 | 24.64 | 0.786 | 0.277 |

100 scenes for training and 8 scenes for testing. To evaluate the generalization ability of our model, we further perform zero-shot evaluation on the synthetic indoor dataset Replica [19], for which we follow [52] to select 8 scenes for testing.

**Implementation Details.** Our FreeSplat is trained end-to-end using Adam [53] optimizer with an initial learning rate of $1e-4$ and cosine decay following [2]. Due to the large GPU requirements of [1, 2] given high-resolution images, all input images are resized to $384 \times 512$ and batch size is set to 1, to form a fair comparison between different methods. We mainly compare with previous generalizable 3DGS methods in 2, 3, 10 reference view settings, where the distance between input views is fixed, thus evaluating the models' performance under different view ranges. For 10 views setting, we also choose target views that are beyond the given sequence of reference views to evaluate the view extrapolation results.

## 5.2 Results on ScanNet

**View Interpolation Results.** On ScanNet, we evaluate the generalizable novel view interpolation results given 2 and 3 reference views as shown in Table 1. Comparing to pixelSplat and MVSplat, our FreeSplat-*spec* consistently improves rendering quality and efficiency on 2-views setting and 3-views setting. Although slightly underperforming on SSIM comparing to NeuRay [9], we show significant improvements on PSNR and LPIPS over NeuRay and $300\times$ faster inference speed. Moreover, our FreeSplat-*fv* consistently offers competitive results given arbitrary number of views, and performs more similarly as FreeSplat-*spec* when number of input views increases.

Table 3: **Novel View Depth Rendering results on ScanNet.** [†]: 10-views results of pixelSplat, MVSplat and FreeSplat-*spec* are given using their 3-views version.

| Method | 2 views | | | 3 views | | | 10 views[†] | | |
| | Abs Diff↓ | Abs Rel↓ | $\delta < 1.25$ ↑ | Abs Diff↓ | Abs Rel↓ | $\delta < 1.25$ ↑ | Abs Diff↓ | Abs Rel↓ | $\delta < 1.25$ ↑ |
|---|---|---|---|---|---|---|---|---|---|
| NeuRay [9] | 0.358 | 0.200 | 0.755 | 0.231 | 0.117 | 0.873 | 0.202 | 0.108 | 0.875 |
| pixelSplat [1] | 1.205 | 0.745 | 0.472 | 0.698 | 0.479 | 0.836 | 0.970 | 0.621 | 0.647 |
| MVSplat [2] | 0.192 | 0.106 | 0.912 | 0.164 | 0.079 | 0.929 | 0.142 | 0.080 | 0.914 |
| **FreeSplat-*spec*** | 0.157 | 0.086 | 0.919 | 0.161 | 0.077 | 0.930 | 0.120 | 0.070 | 0.945 |
| **FreeSplat-*fv*** | 0.153 | 0.085 | 0.923 | 0.162 | 0.077 | 0.928 | 0.097 | 0.059 | 0.961 |

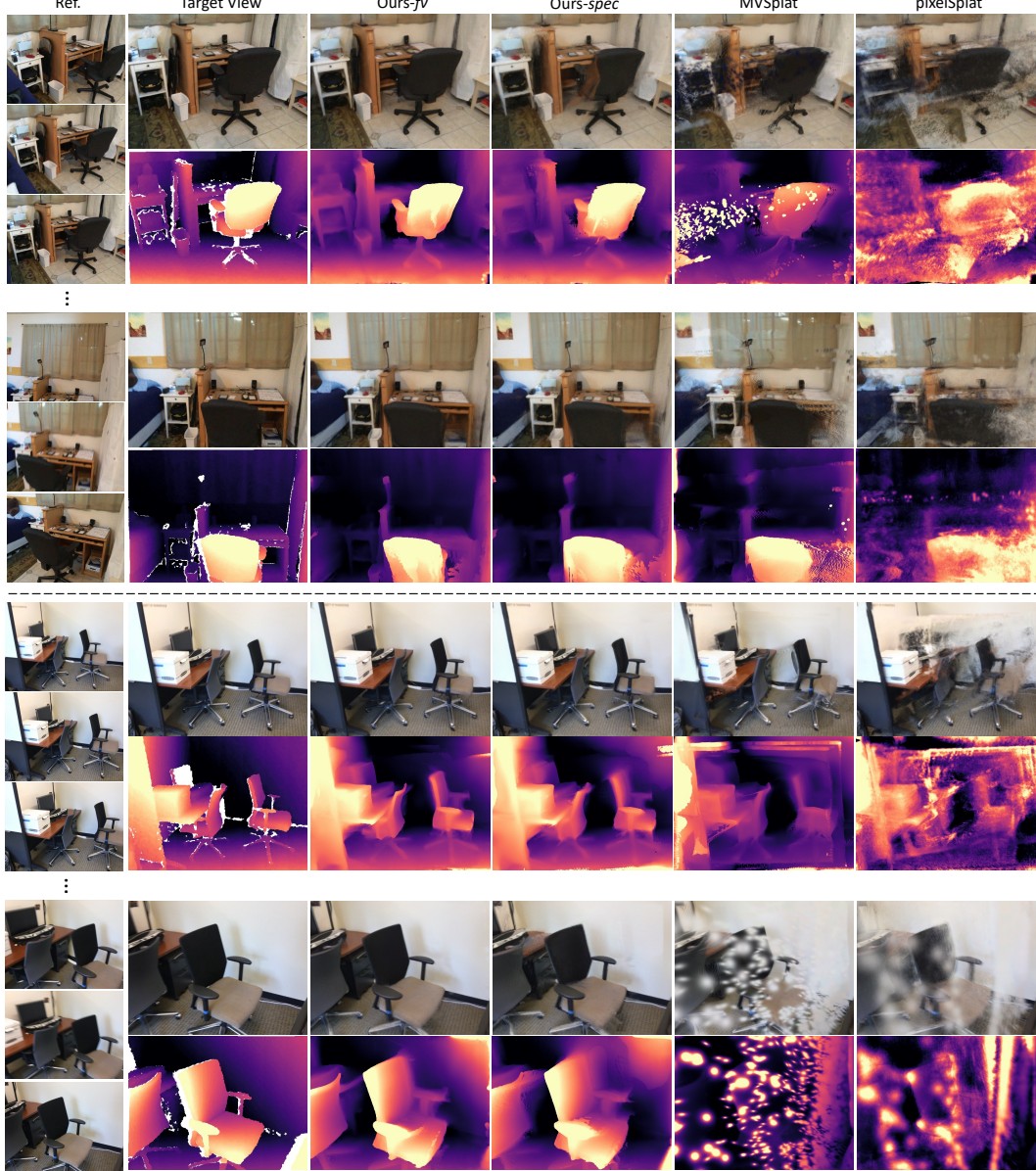

| Ref. | Target View | Ours-*fv* | Ours-*spec* | MVSplat | pixelSplat |

Figure 4: **Qualitative Results of Long Sequence Explicit Reconstruction.** For each sequence, the first two rows are view interpolation results, and the last two rows are view extrapolation results.

**Long Sequence Results.** As shown in Table 2, we further evaluate the long sequence results where we sample reference views with length of 10, and compare both view interpolation and extrapolation results. The results reveal that generalizable 3DGS methods underperform when given long sequence input images, which is due to the complicated camera trajectories in ScanNet, and the inaccuracy of 3D Gaussian localization that leads to errors when observed from wide view ranges. Our FreeSplat-*3views* significantly outperforms pixelSplat and MVSplat on view interpolation and view extrapolation results. Through our proposed FVT that can be easily plugged into our model due to our low requirement on GPU, our FreeSplat-*fv* consistently outperforms our 3-views version. Our PTF module can also reduce the number of Gaussians by around $55.0\%$, which becomes indispensable in long sequence reconstruction due to the pixel-wise unprojection nature of generalizable 3DGS. The qualitative results are shown in Figure 4, which clearly reveal that FreeSplat-*spec* outperforms MVSplat and pixelSplat in localizing 3D Gaussian and preserving fine-grained details, and FreeSplat-*fv* further improves on localizing and fusing multi-view Gaussians.

Table 4: **Zero-Shot Transfer Results on Replica [19].**

| Method | 3 Views | | | | | 10 Views | | | | |
|---|---|---|---|---|---|---|---|---|---|---|
| | PSNR↑ | SSIM↑ | LPIPS↓ | $\delta < 1.25$ ↑ | #GS(k) | PSNR↑ | SSIM↑ | LPIPS↓ | $\delta < 1.25$ ↑ | #GS(k) |
| pixelSplat [1] | 26.24 | 0.829 | 0.229 | 0.576 | 1769 | 19.23 | 0.719 | 0.414 | 0.375 | 5898 |
| MVSplat [2] | 26.16 | 0.840 | 0.173 | 0.670 | 590 | 18.66 | 0.717 | 0.360 | 0.565 | 1966 |
| **FreeSplat-*spec*** | 26.98 | 0.848 | 0.171 | 0.682 | 423 | 21.11 | 0.762 | 0.312 | 0.720 | 1342 |
| **FreeSplat-*fv*** | 26.64 | 0.843 | 0.184 | 0.682 | 421 | 21.95 | 0.777 | 0.290 | 0.742 | 1346 |

Table 5: **Ablation on ScanNet.** CV: Cost Volume, PTF: Pixel-wise Triplet Fusion, FVT: Free-View Training.

| CV | PTF | FVT | 3 views | | | | | 10 views | | | | |
|---|---|---|---|---|---|---|---|---|---|---|---|---|
| | | | PSNR↑ | SSIM↑ | LPIPS↓ | $\delta < 1.25$ ↑ | $\delta < 1.10$ ↑ | PSNR↑ | SSIM↑ | LPIPS↓ | $\delta < 1.25$ ↑ | $\delta < 1.10$ ↑ |
| ✓ | | | 27.12 | 0.825 | 0.224 | 0.925 | 0.762 | 24.23 | 0.792 | 0.277 | 0.942 | 0.804 |
| | ✓ | | 22.10 | 0.696 | 0.359 | 0.639 | 0.311 | 17.94 | 0.607 | 0.487 | 0.543 | 0.216 |
| ✓ | ✓ | | 27.45 | 0.829 | 0.222 | 0.930 | 0.773 | 25.15 | 0.800 | 0.278 | 0.945 | 0.823 |
| ✓ | | ✓ | 26.41 | 0.806 | 0.232 | 0.919 | 0.746 | 25.40 | 0.799 | 0.252 | 0.950 | 0.831 |
| ✓ | ✓ | ✓ | 27.34 | 0.826 | 0.226 | 0.928 | 0.764 | 25.90 | 0.808 | 0.252 | 0.961 | 0.858 |

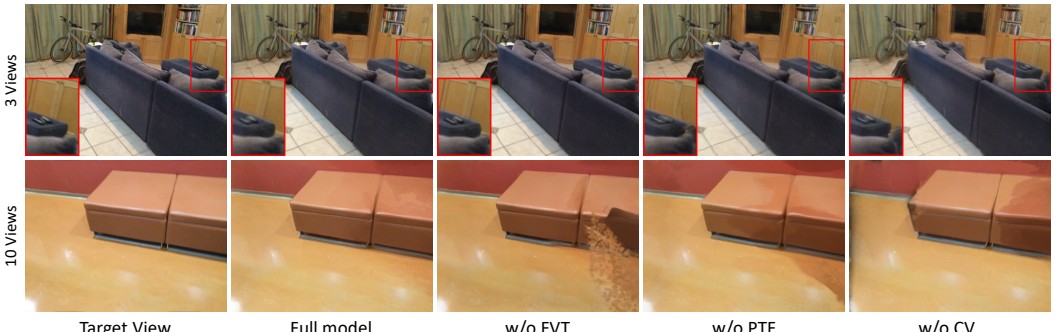

Figure 5: **Qualtitative Ablation Study.** The first and second row use input view lengths of 3 and 10.

**Novel View Depth Estimation Results.** We also investigate the correctness of 3D Gaussian localization of different methods through comparing their depth rendering results. We report the Absolute Difference (Abs. Diff), Relative Difference (Rel. Diff), and threshold tolerance $\delta < 1.25$ results from novel views in Table 3. We find that FreeSplat consistently outperforms pixelSplat and MVSplat in predicting accurately localized 3D Gaussians, where FreeSplat-*fv* reaches 94.9% of $\delta < 1.25$, enabling accurate unsupervised depth estimation on novel views. The improved depth estimation accuracy of FreeSplat-*fv* highlights the importance of depth estimation in supporting free-view synthesis across broader view range.

### 5.3 Zero-Shot Transfer Results on Replica

We further evaluate the zero-shot transfer results through testing on Replica dataset, with results in Table 4. Our view interpolation and novel view depth estimation results still outperforms existing methods. The long sequence results degrade due to inaccurate depth estimation and domain gap, indicating potential future work in further improving the depth estimation in zero-shot tranferring.

### 5.4 Ablation Study

We conduct a detailed ablation study as shown in Table 5 and Figure 5. The results indicate that: 1) cost volume is essential in accurately localizing 3D Gaussians; 2) our proposed PTF module can consistently contribute to rendering quality and depth estimation results. The PTF module learns to incrementally fuse multi-view 3D Gaussians and contributes significantly when varying number of input views, and serves as a multi-view localization regularization that helps unsupervised depth estimation; 3) Our FVT module excels in long sequence reconstruction quality as well as novel view

depth rendering results, which provides stricter constrains on 3D Gaussian localization and can be seamlessly combined with the PTF module to fit to varying length of input views.

## 6 Conclusion

In this study, we introduced FreeSplat, a generalizable 3DGS model that is tailored to accommodate an arbitrary number of input views and perform free-view synthesis using the global 3D Gaussians. We developed a Low-cost Cross-View Aggregation pipeline that enhances the model's ability to efficiently process long input sequences, thus incorporating stricter geometry constraints. Additionally, we have devised a Pixel-wise Triplet Fusion module that effectively reduces redundant pixel-aligned 3D Gaussians in overlapping regions and merges multi-view Gaussian latent features. FreeSplat consistently improves the fidelity of novel view renderings in terms of both color image quality and depth map accuracy, facilitating feed-forward global Gaussians reconstruction without depth priors.

## 7 Acknowledgement

This work is supported by the Agency for Science, Technology and Research (A*STAR) under its MTC Programmatic Funds (Grant No. M23L7b0021).

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

# A Appendix / supplemental material

## A.1 Experimental Environment

We conduct all the experiments on single NVIDIA RTX A6000 GPU. The experimental environment is PyTorch 2.1.2 and CUDA 12.2.

## A.2 Additional Implementation Details

We set the number of virtual depth planes $K = 128$, matching feature dimension $C_m = 64$, and $d_{near} = 0.5, d_{far} = 15.0$ for cost volume formulation, and set $\delta = 0.05$ in Eq.(8) for pixel-wise alignment. To train the 3-views version of pixelSplat [1] on a single NVIDIA RTX A6000 GPU, we change their ViT patch size from $8 \times 8$ to $16 \times 16$. During inference on the 10 views setting, the epipolar line sampling in pixelSplat and the cross-view attention in MVSplat [2] are performed between nearby views similarly as ours to save GPU requirements and form a fair comparison. For the free-view version of FreeSplat we set the number of nearby views as $N = 4$ for training. For the testing of long sequence explicit reconstruction, cost volumes are formed between nearby 8 views.

## A.3 Additional Experiments

Table 6: **Comparison on computational cost and whole scene reconstruction (30 input views).** We report the required GPU for Train / Test, the Encoding Time, the rendering FPS, and PSNR of novel views. - denotes that we are not able to run pixelSplat inference using 30 input views due to its increasing GPU requirement.

| Method | GPU (GB) | Time (s)↓ | FPS↑ | PSNR↑ |
|---|---|---|---|---|
| pixelSplat-3views | 44.9 / - | - | - | - |
| MVSplat-3views | 34.8 / 44.0 | 3.004 | 39 | 17.57 |
| **FreeSplat-*3views*** | 16.9 / 21.0 | 1.191 | 57 | 21.33 |
| **FreeSplat-*fv* w/o PTF** | 42.2 / 21.0 | **1.006** | 39 | 21.82 |
| **FreeSplat-*fv*** | 42.2 / 21.0 | 1.205 | **72** | **22.32** |

**Computational Cost.** As shown in Table 6, we compare the required GPU memory for training and testing, the encoding time, rendering FPS, and PSNR for whole scene reconstruction. pixelSplat-3views and MVSplat-3views already consume 30 50 GB GPU memory for training due to their quadratically increasing GPU memory requirement with respect to the image resolution / sequence length. Therefore, it becomes infeasible to extend their methods to higher resolution inputs or longer sequence training. In comparison, our low-cost framework design enable us to effectively train on long sequence inputs while requiring lesser GPU memory compared to the 3 views version of existing methods. Furthermore, our proposed PTF module can effectively reduce redundant 3D Gaussians, improving rendering speed from 39 to 72 FPS . This becomes increasingly important when reconstructing larger scenes since generalizable 3DGS normally perform pixel-wise unprojection, which can easily result in redundancy in the overlapping regions.

Table 7: **Results on RE10K and ACID with 2 input views.** We train our model on RE10K, and report its results on RE10K and ACID. * denotes that our model is trained on our previously downloaded 9,266 scenes instead of 11,075 scenes used by baselines.

| Method | RE10K - 2 Views | | | ACID - 2 Views | | |
|---|---|---|---|---|---|---|
| | PSNR↑ | SSIM↑ | LPIPS↓ | PSNR↑ | SSIM↑ | LPIPS↓ |
| pixelSplat | 25.89 | 0.858 | 0.142 | 27.64 | 0.830 | 0.160 |
| MVSplat | 26.39 | 0.869 | **0.128** | **28.15** | **0.841** | **0.147** |
| **Ours*** | **26.41** | **0.871** | 0.132 | 27.94 | 0.838 | 0.157 |

**Experiments on RE10K and ACID.** To further evaluate our model's generalization ability across diverse domains, we train our model on RE10K using 2-View setting and 5-View setting respectively. The results are shown in Table 7, 8 and Figure 6. Note that for the 5-View setting inference, we sample input views with random intervals between 25 and 45 due to the limited sequence lengths in RE10K and ACID. In the 2-View setting, we perform better than pixelSplat and on par as MVSplat on both datasets. In the 5-View setting, we outperform both baselines by a clear margin. We analyze the main causes of the above results as follows:

Table 8: **Results on RE10K and ACID with 5 input views.** The models are trained on RE10K with 5 input views.

| Method | RE10K - 5 Views | | | ACID - 5 Views | | |
| --- | --- | --- | --- | --- | --- | --- |
| | PSNR↑ | SSIM↑ | LPIPS↓ | PSNR↑ | SSIM↑ | LPIPS↓ |
| pixelSplat | 24.78 | 0.850 | 0.150 | 26.84 | 0.833 | 0.173 |
| MVSplat | 25.38 | 0.866 | 0.132 | 27.81 | 0.863 | 0.134 |
| **Ours** | **25.95** | **0.873** | **0.128** | **28.35** | **0.870** | **0.130** |

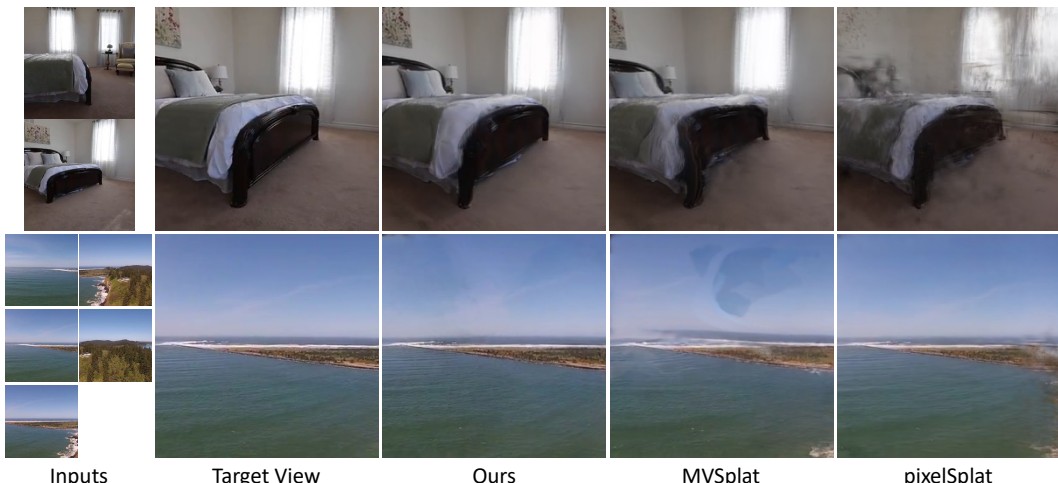

Figure 6: **Qualitative Results on RE10K and ACID.** We visualize the 2-Views results on RE10K and 5-Views results on ACID.

In the 2-view comparison experiments with the baselines, the image interval between the given stereo images were set to be large. On average, the interval between image stereo is 66 in RE10K and 74 in ACID, which is much larger than our indoor datasets setting (20 for ScanNet and 10 for Replica). Such large interval can result in minimum view overlap between the image stereo, which means that our cost volume can be much sparser and multi-view information aggregation is weakened. In contrast, MVSplat uses a cross-view attention that aggregates multi-view features through a sliding window which does not leverage camera poses. pixelSplat uses a heavy 2D backbone that can potentially become stronger monocular depth estimator. In our 5-view setting, we outperform both baselines by clear margins. This is partially due to the smaller image interval and larger view overlap between nearby views. As a result, our cost volume can effectively aggregate multi-view information, and our PTF module can perform point-level fusion and remove those redundant 3D Gaussians.

Therefore, our model is not specifically designed for highly sparse view inputs, but it is designed as a low-cost model that can easily take in much longer sequences of higher-resolution inputs, that is suitable for indoor scene reconstruction. Comparing to RE10K and ACID, real-world indoor scene sequences usually contain more complicated camera rotations and translations, which results in the requirement of more dense observations to reconstruct the 3D scenes with high completeness and accurate geometry. Consequently, our model is targeting the fast indoor scene reconstruction with keyframe inputs, which contain long sequences of high-resolution images, while existing works struggle to extend to such setting as evaluated in our main paper.

**Comparison with SurfelNeRF.** We further compare with SurfelNeRF as shown in Table 9 and Figure 7. We evaluate on the same novel views as theirs, sampling input views along their input sequences with an interval of 20 between nearby views. Note that the number of input views changes when the input length changes, while our FreeSplat-fv can seamlessly conduct inference with arbitrary numbers of inputs. Our method performs significantly better than SurfelNeRF in both rendering quality and efficiency. Our end-to-end framework jointly learns depths and 3DGS using an MVS-based backbone, while SurfelNeRF relies on depths and does not aggregate multi-view features to assist their surfel feature prediction.

Table 9: **Comparison with SurfelNeRF.** We compare with SurfelNeRF on the same sequences as their test set. * denotes the rendering speed is reported from SurfelNeRF.

| Method | Time (s) | FPS↑ | PSNR↑ | SSIM↑ | LPIPS↓ |
|---|---|---|---|---|---|
| SurfelNeRF | 3.242 | 5* | 24.20 | 0.694 | 0.477 |
| **FreeSplat-*fv*** | **0.302** | **224** | **27.06** | **0.818** | **0.223** |

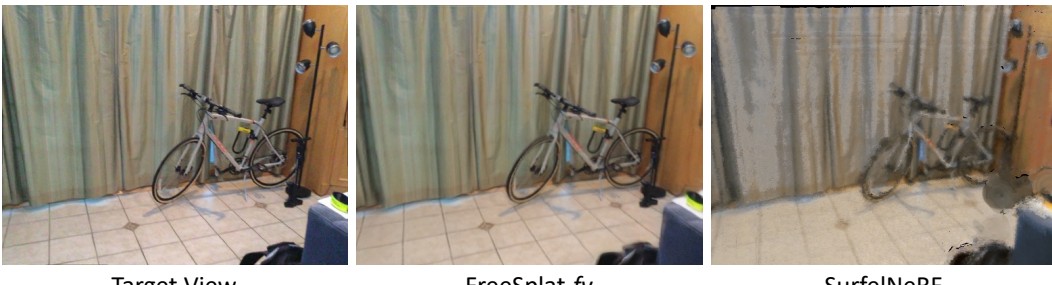

| Target View | FreeSplat-*fv* | SurfelNeRF |
|---|---|---|

Figure 7: **Qualitative Comparison with SurfelNeRF.**

## A.4 Additional Qualitative Results

**2 and 3-View Interpolation Results.** The qualitative results are shown in Figure 8, where FreeSplat more precisely localizes 3D Gaussians and captures more fine-grained details comparing to previous methods. FreeSplat can also localize 3D Gaussians more accurately and renders precise depth maps, supporting high-quality rendering from broader view range (*cf.* FreeSplat-*spec* results in Figure 4).

**Results on Replica.** We show the qualitative results on Replica in Figure 10, where our superiority over MVSplat and pixelSplat remains. The results indicate the generalization ability of FreeSplat across indoor datasets for the view interpolation task.

**Results of Whole Scene Reconstruction.** We also show qualitative results of our whole scene reconstruction in Figure 9. Despite the long input sequence ($\sim 40$ images) covering the whole scene, FreeSplat can still perform efficient feed-forward in $\sim 1$s on single NVIDIA RTX A6000, and can render high-quality images and accurate depth maps from novel views. On the other hand, it is still difficult to accurately predict depth of textureless (*e.g.* wall) and specular (*e.g.* light reflection on the floor) regions. However, we hope our work provides an initial step towards accurate geometry reconstruction without ground truth depth priors.

## A.5 Limitations

Although our approach excels in novel view rendering depth estimation and support arbitrary number of input views, the GPU requirement becomes expensive ($> 40$GB) when inputting extremely long image sequence ($> 50$). On the other hand, due to our unsupervised scheme of depth estimation, there is still a gap between our 3D reconstruction accuracy and the state-of-the-art methods with 3D supervision [33, 32] or RGB-D inputs [36, 35] (*e.g.* as shown in Figure 9, the textureless and specular regions). Our main focus is to explore the feed-forward indoor scene photorealistic reconstruction purely based on 2D supervision.

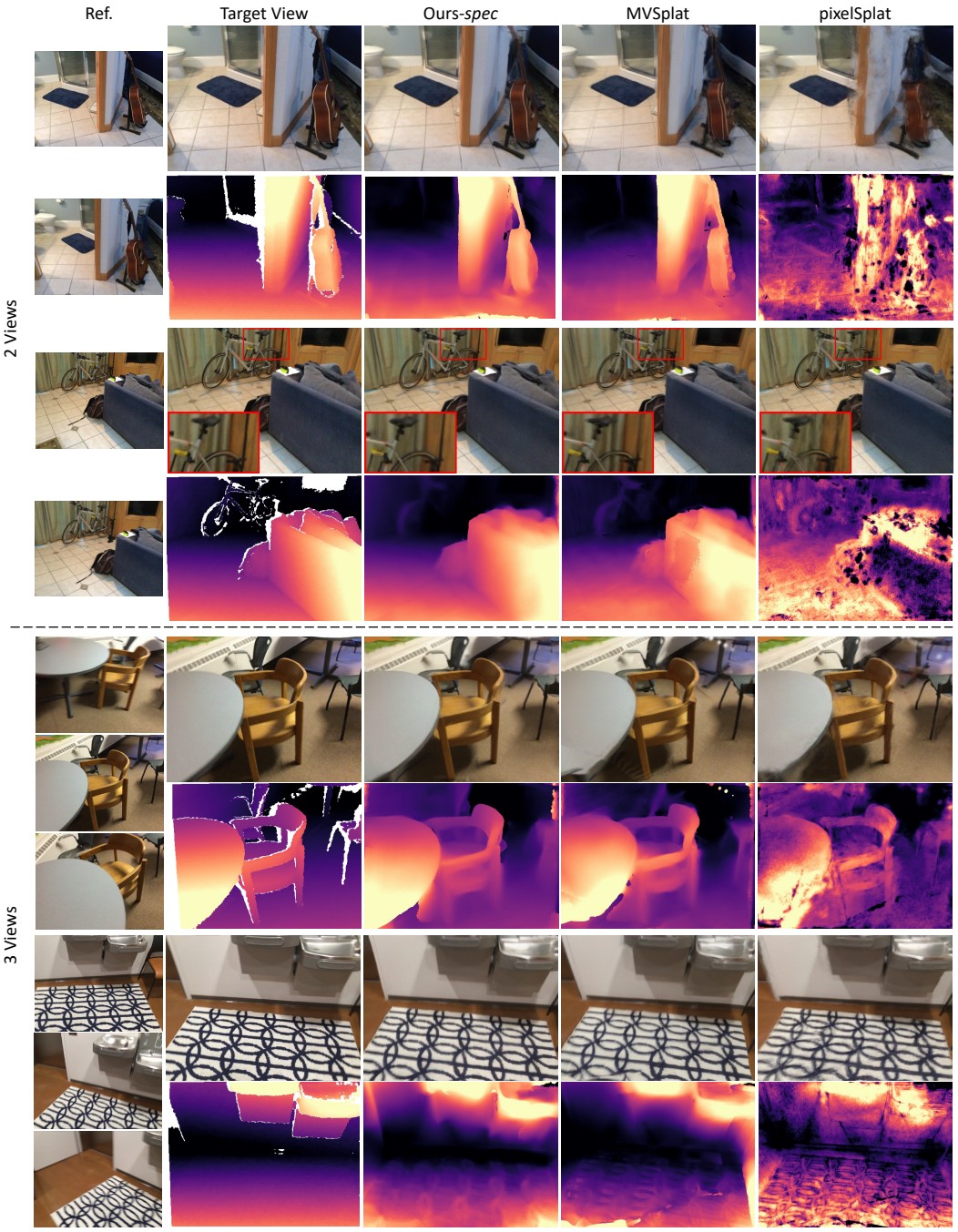

Figure 8: **Qualtitative Results given 2 and 3 reference views.** We show the rendered color images (first row) and depth maps (second row) for each batch of reference views.

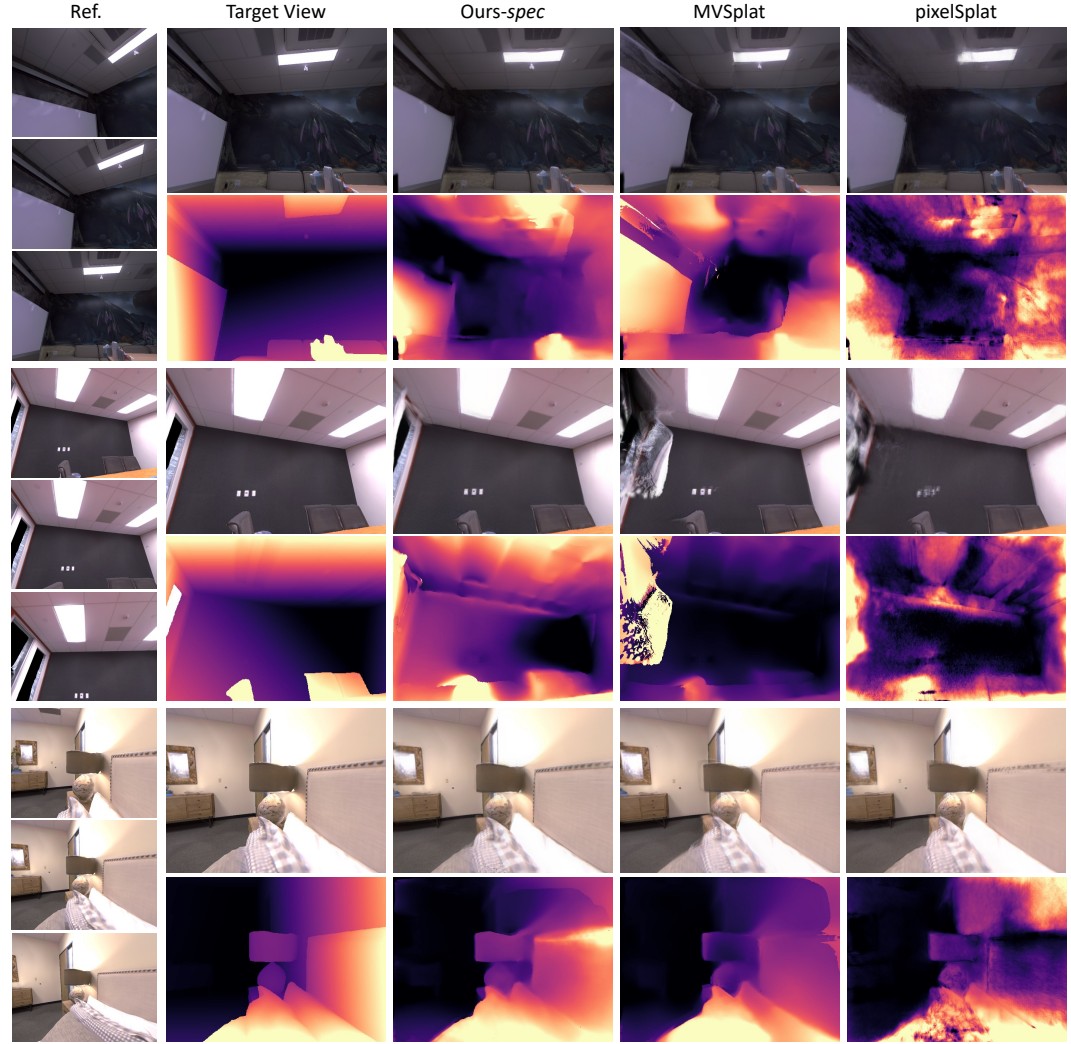

Figure 9: **Qualitative Results on Replica.**

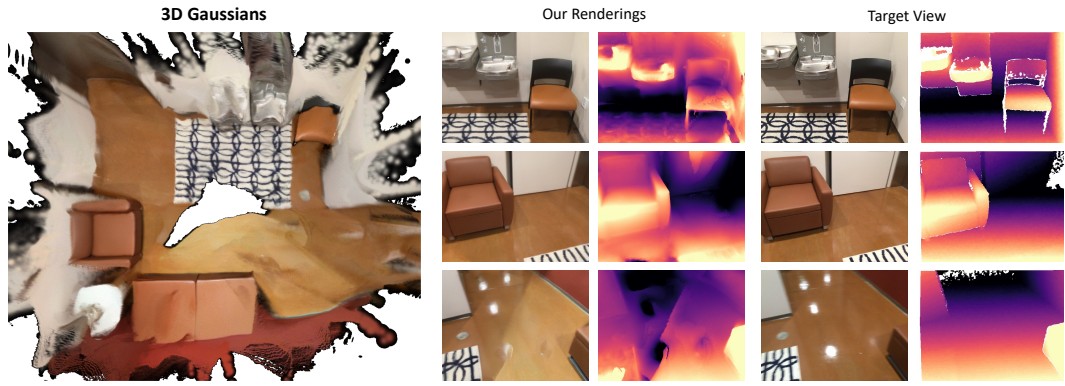

Figure 10: **Qualitative Results of whole scene reconstruction.**

