# OpenReview forum: "FreeSplat: Generalizable 3D Gaussian Splatting Towards Free View Synthesis of Indoor Scenes"
_NeurIPS.cc/2024/Conference — NeurIPS 2024 poster_

### Official Review · Reviewer_ZuRj · 2024-06-28

**Soundness:** 2
**Presentation:** 2
**Contribution:** 2
**Rating:** 5
**Confidence:** 3

**Summary:**

This paper introduces a generalizable 3DGS model which is capable of reconstructing geometrically consistent 3D scenes from long sequence input towards free-view synthesis. The key idea of this paper is introducing Low-cost Cross-View Aggregation, which makes it is possible to use more nearby views for feature matching. Experiments show the advance in long sequence novel view synthesis.

**Strengths:**

1. Low-cost Cross-View Aggregation is introduced to predict initial Gaussian triplets, with which the computational cost would be lower, making it possible for feature matching between more nearby views and training on long sequence reconstruction.
2. The Gaussian triplets are fused by Pixel-wise Triplet Fusion. This module can effectively reduce the pixel-aligned Gaussian redundancy in the overlapping regions and aggregate multi-view 3D Gaussian latent features.
3. The experiments show the advance in long sequence novel view synthesis.

**Weaknesses:**

1. It would be better if the authors could report the training time of their model, since the authors addressed that both the Low-cost Cross-View Aggregation module and Pixel-wise Triplet Fusion module can lower the computation cost.
2. It would be better to include the comparison with SurfelNeRF, since using a depth estimator differs from the GT depth.
3. The definition of FreeSplat-fv and  FreeSplat-spec is not given clearly in this paper.

**Questions:**

1. I want to make it clear that FreeSplat-fv indicates using Free-View Training (FVT) strategy and FreeSplat-spec indicates using specific number of reference views, such as two or three.
2. Gaussian Splatting SLAM is one of the 3DGS-based SLAM system, which also can be run without depth as input information.It would be better if the authors can explain the strength of their work compared with this kind of 3DGS-based RGB SLAM, since the SLAM method can run in real time and process a longer input sequence.

**Limitations:**

Yes. As the authors discussed in this paper, the GPU requirement becomes expensive (> 40GB) when inputting extremely long image sequence (> 50). Besides, there is still a gap between the method of this paper and the other methods that rely on depth information as input.

---

> ### Author Rebuttal · Authors · 2024-08-06
>
> ## To Reviewer ZuRj (#R4):
>
> 1. **Computational Cost:** The full training time is around **2 days** for our 2-view and 3-view  versions of our method and baselines, and **3 days** for our free-view version. Although our training time is similar to pixelSplat [1] and MVSplat [2], we consume **much fewer GPU hours** due to our lower GPU requirements. As shown in our ***rebuttal pdf Table 1***, the 3-view version of pixelSplat and MVSplat already consume **30~50 GB** for training. It is due to their heavy patch-based 2D transformer and cross-view attention that results in quadratically increasing GPU requirements with increased image resolution and input sequence length. Therefore, their models are not suitable for high-resolution, long sequence inputs. In contrast, our proposed low-cost backbone can greatly reduce GPU consumption given a long sequence of high-resolution inputs, making whole scene reconstruction much more feasible. Furthermore, our proposed PTF module can effectively reduce redundant 3D Gaussians, improving the rendering speed from 39 to 72 FPS in whole scene reconstruction, which becomes even more important when reconstructing larger scenes to achieve real-time rendering. Overall, the efficiency of our model mainly includes: (1) lowering required GPU to make it feasible to train on long sequences of high-resolution inputs, and also reducing training cost (required GPU hours); (2) removing redundant 3D Gaussians to increase rendering speed and handle large 3D scenes.
>
> 2. **Comparison with SurfelNeRF:** As shown in our ***rebuttal pdf Table 3 and Figure 4***, we conduct a comparison experiment with SurfelNeRF. We evaluate our FreeSplat-*fv* on the same novel views as theirs. Note that their input views are much denser than ours: e.g. they input 40 views covering a length of 120 images, with an approximate image interval of 3; while we still use relatively sparse inputs that have a fixed image interval of 20. We pick input views along their image sequences and evaluate at the same novel views. Note that the number of input views changes with the length of image sequence, while our FreeSplat-*fv* can perfectly fit to such varying numbers of input views. We outperform them largely in both rendering quality and efficiency. We analyze that such improvements are creditable to two main reasons:
>
>    (1) Their surfel-based representation is less representative comparing to 3DGS;
>
>    (2) Their model is not trained end-to-end with MVS encoder. SurfelNeRF uses external GT depth maps or MVS depth maps and do not jointly train an MVS-based encoder with the surfel-predictor, so their surfel-predictor does not have multi-view information, resulting in sub-optimal rendering quality.
>
> 3. **Definition of FreeSplat-spec and FreeSplat-fv:** You are correct that FreeSplat-spec is trained using fixed number of input views for a fair comparison with the baselines FreeSplat-fv uses Free-View Training to reconstruct larger regions for stricter supervision from broader view range, to enforce precise 3D Gaussian localization. We will add the above illustration in the final version to improve the clarity.
>
> 4. **Comparison with 3DGS-based SLAM methods:** Comparing to 3DGS-based SLAM methods, our proposed generalizable 3DGS-based method has the following two main strengths:
>
>    **(1) Efficiency:** Although the existing 3DGS-based SLAM can reach *1~3* FPS for per-scene optimization, they still require *10~20* minutes to finish reconstructing the whole 3D scene (*cf.* MonoGS [3] paper Table 9, 10; Splatam [4] paper Table 6). In contrast, our method only requires *1~2* seconds to parallelly mapping all the input views to 3D Gaussians. Furthermore, our proposed PTF can largely improve the rendering FPS (*cf.* our ***rebuttal pdf Table 1***), making the scene-level generalizable 3DGS more feasible. After fast feeding-forward, we can optionally conduct fast per-scene optimization using our predicted 3D Gaussians as the initalization to further improve the rendering quality.
>
>    **(2) Less constraints on Input Data:** Majority of the existing SLAM methods require dense input RGB-D sequence to estimate camera trajectories on-the-fly. Although several works [3, 5] can handle monocular inputs, there is still a gap between their camera tracking accuracy and RGB-D methods (*cf.* MonoGS paper Table 1). Such on-the-fly methods may also suffer from the drift problem due to accumulated errors. In contrast, our method is designed to explore learning strong priors from training data and perform fast offline prediction instead of on-the-fly application, making it more compatible with COLMAP and recent feed-forward pose estimation methods like DUSt3R [6]. Furthermore, our method does not require sensor depths during training or inference, and only requires relatively sparse inputs, which can be more easily generalized to various domains.
>
>
>
> [1] Charatan, David, et al. "pixelsplat: 3d gaussian splats from image pairs for scalable generalizable 3d reconstruction." Proceedings of the IEEE/CVF Conference on Computer Vision and Pattern Recognition. 2024.
>
> [2] Chen, Yuedong, et al. "Mvsplat: Efficient 3d gaussian splatting from sparse multi-view images." arXiv preprint arXiv:2403.14627 (2024)."
>
> [3] Matsuki, Hidenobu, et al. "Gaussian splatting slam." *Proceedings of the IEEE/CVF Conference on Computer Vision and Pattern Recognition*. 2024.
>
> [4] Keetha, Nikhil, et al. "SplaTAM: Splat Track & Map 3D Gaussians for Dense RGB-D SLAM." *Proceedings of the IEEE/CVF Conference on Computer Vision and Pattern Recognition*. 2024.
>
> [5] Teed, Zachary, and Jia Deng. "Droid-slam: Deep visual slam for monocular, stereo, and rgb-d cameras." *Advances in neural information processing systems* 34 (2021): 16558-16569.
>
> [6] Wang, Shuzhe, et al. "Dust3r: Geometric 3d vision made easy." *Proceedings of the IEEE/CVF Conference on Computer Vision and Pattern Recognition*. 2024.

---

> > ### Comment · Reviewer_ZuRj · 2024-08-11
> >
> > I appreciate the authors for their detailed rebuttal, which addresses my questions. If the authors can include the comparison with 3DGS-based SLAM methods in their paper, it would be better to understand their method's strengths. I will keep my initial rating.

---

> ### Author Response · Authors · 2024-08-08
> **Correct an error in our rebuttal**
>
> In my previous rebuttal on the **comparison with SurfelNeRF**, we erroneously referred to ***rebuttal pdf Table 3 and Figure 4*** when discussing the experimental results. We would like to clarify that the correct reference should be to ***rebuttal pdf Table 4 and Figure 5***. We apologize for any confusion this may have caused and appreciate the opportunity to correct this error.

---

> ### Author Response · Authors · 2024-08-12
>
> Thank you for your insightful feedback. To provide a more thorough comparison with 3DGS-based SLAM methods utilizing monocular inputs, we reproduced the monocular version of MonoGS (referred to as MonoGS-Mono) and conducted experiments on the `scene0316_00` from the ScanNet dataset. We compared these results against our **whole scene reconstruction** approach. To ensure a fair comparison, we present their results both with and without ground truth (GT) camera poses. The results are summarized in the table below:
>
> |     Method     | GT Poses | ATE RMSE (m) | Time (s)↓ |   PSNR↑   |   SSIM↑   |  LPIPS↓   |
> | :------------: | :------: | :-: | :-----: | :-------: | :-------: | :-------: |
> |  MonoGS-Mono   |          |    0.2615    |   574.8   |   20.18   | **0.830** |   0.459   |
> |  MonoGS-Mono   |    ✔     |      -       |   347.4   |   16.50   |   0.756   |   0.533   |
> | FreeSplat-*fv* |    ✔     |      -       |  **1.2**  | **22.17** |   0.818   | **0.313** |
>
> The results demonstrate that, on the real-world ScanNet dataset, MonoGS struggles to accurately track the camera trajectory and reconstruct a geometrically correct 3D scene with only color images as the input. The predicted camera trajectory suffers from significant deviation due to the drift problem and the lack of depth priors / sensor depth inputs. Interestingly, providing ground truth camera poses results in a notable performance drop, which we attribute to the difficulty of fitting reconstructed 3D Gaussians to the training frames without pose optimization. In contrast, our method effectively learns geometry priors from the training data and performs significantly faster feed-forward predictions. Our approach also significantly outperforms MonoGS-Mono in rendering quality, particularly in terms of PSNR and LPIPS metrics. These results demonstrate the advantages of our generalizable 3DGS-based method for whole scene reconstruction when using monocular color inputs, offering both effectiveness and efficiency.
>
> We sincerely appreciate your comments, which prompted us to perform this comparison with 3DGS-based SLAM methods using monocular inputs. This experiment further solidifies our belief in the unique strengths of our approach as a foundational step toward feed-forward 3DGS-based whole scene reconstruction. We hope the above experiment and analysis can adequately address your concerns about the comparison with 3DGS-based SLAM methods.

---

> > ### Comment · Reviewer_ZuRj · 2024-08-12
> >
> > Thanks for this comparison. This is an interesting and more clear result, which can show your strength in scene reconstruction when given GT cam poses compared to the 3DGS-based SLAM method.

---

### Official Review · Reviewer_Hd37 · 2024-07-05

**Soundness:** 3
**Presentation:** 3
**Contribution:** 3
**Rating:** 6
**Confidence:** 4

**Summary:**

This paper proposes a FreeSplat, aiming at generalizable 3D gaussian splitting for long sequence inputs. Specifically, it uses an efficient CNN-based cost volume and eliminates redundant 3D gaussians observed across multiple views. Extensive experiments show that FreeSplat effectively reduces inference costs and improves the novel view quality for long sequence inputs.

**Strengths:**

* The writing is good, making it easy to understand and follow.

* The proposed Pixel-Aligned Triplet Fusion (PTF) is interesting as it effectively handles the redundant 3D gaussians for long sequence inputs.

**Weaknesses:**

* I think the evaluation details are not clear. For example, during the evaluation, I am curious whether the maximum gap between input views is fixed regardless of the number of input views. Does "view range" increase as the number of input views increases? If so, please describe the evaluation details.

* It is weird that the novel view image quality drops as the number of input views increases, but the novel view depth quality increases. I'd like more clarification on this.

* The existing pixelSplat and MVSplat have been experimented with on RealEstate10k [1] and ACID [2], and both datasets also contain long sequence inputs. The authors should have included comparisons on at least one of the RealEstate10k and ACID datasets.

[1] Zhou et al., Stereo magnification: Learning view synthesis using multiplane images, SIGGRAPH 2018.

[2] Liu et al., Infinite nature: Perpetual view generation of natural scenes from a single image, ICCV 2021.

**Questions:**

The proposed method is interesting, but the lack of evaluation details, lack of comparative experiments, and inconsistencies in the results make it difficult to verify the contribution. The authors should address the "weaknesses" and strengthen the manuscript.

---

> ### Author Rebuttal · Authors · 2024-08-06
>
> ## To Reviewer Hd37 (#R3):
>
> 1. **Evaluation Details:** For view range, as mentioned in Line \#431-432 in the appendix, the distance between nearby input views is fixed to 20 in ScanNet and 10 in Replica, thus the maximum gap increases linearly with the number of input views. Therefore, the long sequence reconstruction covers larger regions of the scene and supports whole scene reconstruction. During evaluation, we randomly select a sequence of images with specific length (2, 3, 10) and fixed interval between nearby views, e.g. we choose the 30, 50, 70-th images of a ScanNet scene as input views for the 3-view setting. When evaluating interpolation results, we randomly choose novel views between each interval of nearby views, and for extrapolation we choose novel views that are beyond the input image sequence as illustrated in Line \#226-227 in our main paper. We will include the above illustrations in the final version to describe the evaluation details more clearly.
>
> 2. **Decreased image quality when the number of input views increases:** It is because of the increased difficulty when explicitly reconstructing longer sequences. Similarly as existing works, when evaluating interpolation results we select novel views that are within the view interval between nearby views. When using 2 or 3 input views, and the reconstructed 3D Gaussians only needs to give high-quality renderings from a relatively narrow view range, which means that they may still give good renderings even if the 3D Gaussians are erroneously located. However, when increasing the number of input views to 10, the reconstructed region needs to give reasonable renderings from a broader view range, where the distant novel views can serve as extrapolation views, thus evaluating the precision of Gaussian localisation. For example, given 10 input views of the 10, 30, 50, 70, 90, 110, 130, 150, 170, 190-th images of the scene, we render novel view from the pose of the 185-th image, then it may view at the same region as in the 10-th and 30-th images but from a significantly differently view direction. Such a case requires the reconstructed 3D Gaussians to be precisely localized, otherwise they may become floaters when viewing from extrapolated views. On the other hand, the depth map quality only evaluates the localization accuracy of 3D Gaussians, which becomes more precise when given more reference views and training on long sequences. Overall, training the model on long sequences is important for precise 3D Gaussian localization since the distant novel views can serve as extrapolated views to regularize the depth estimation, which is one of the key focuses of our paper.
>
> 3. **Experiments on Re10k and ACID:**  To further evaluate our model's generalization ability across diverse domains, we train our model on RE10K using 2-View setting and 5-View setting respectively. The results are shown in our ***rebuttal pdf Table 2, 3 and Figure 3***. Note that for the 5-View setting inference, we sample input views with random intervals between 25 and 45 due to the limited sequence lengths in RE10K and ACID. In the 2-View setting, we perform better than pixelSplat [1] and on par as MVSplat [2] on both datasets. In the 5-View setting, we outperform both baselines by a clear margin. We analyze the main causes of the above results as follows:
>
>    In the 2-view comparison experiments with the baselines, the image interval between the given stereo images were set to be large. On average, the interval between image stereo is 66 in RE10K and 74 in ACID, which is much larger than our indoor datasets setting (20 for ScanNet and 10 for Replica). Such large interval can result in **minimum view overlap** between the image stereo (e.g. as shown in our ***rebuttal pdf Figure 4(b)***), which means that our cost volume can be **much sparser** and multi-view information aggregation is weakened. In contrast, MVSplat uses a cross-view attention that aggregates multi-view features through a sliding window which does not leverage camera poses. pixelSplat uses a heavy 2D backbone that can potentially become stronger monocular depth estimator. In our 5-view setting, we outperform both baselines by clear margins. This is partially due to the smaller image interval and larger view overlap between nearby views. As a result, our cost volume can effectively aggregate multi-view information, and our PTF module can perform point-level fusion and remove those redundant 3D Gaussians.
>
>    Therefore, our model is not specifically designed for highly sparse view inputs, but it is designed as a low-cost model that can easily take in much longer sequences of higher-resolution inputs, that is suitable for indoor scene reconstruction (we also offer a quantitative comparison on computation cost in our ***rebuttal Table 1*** to emphasize our strengths). Comparing to RE10K and ACID, real-world indoor scene sequences usually contain more complicated camera rotations and translations, which results in the requirement of more dense observations to reconstruct the 3D scenes with high completeness and accurate geometry. Consequently, our model is targeting the fast indoor scene reconstruction with keyframe inputs, which contain long sequences of high-resolution images, while existing works struggle to extend to such setting as evaluated in our main paper.
>
>    We really appreciate your question, which helped us dive deeper into broader experimental comparisons and explore the underlying reasons, reaching a better clarification of our research focus and contributions. We will add the corresponding results and illustrations in final version to improve the completeness and highlight the contributions of our paper.
>
> [1] Charatan, David, et al. "pixelsplat: 3d gaussian splats from image pairs for scalable generalizable 3d reconstruction." CVPR. 2024.
>
> [2] Chen, Yuedong, et al. "Mvsplat: Efficient 3d gaussian splatting from sparse multi-view images." arXiv preprint arXiv:2403.14627 (2024)."

---

> > ### Comment · Reviewer_Hd37 · 2024-08-08
> >
> > Thanks for the response. It well addressed my concerns for **Evaluation Details** and **Experiments on Re10k & ACID**.
> > However, I need more clarification on **Decreased image quality when the number of input views increases**.
> >
> > I understand that fusing redundant gaussians can lead to more accurate gaussian localization, but I'm still confused why image quality seems to decrease as the number of input views increases. In the current experimental setup, making a fair comparison is challenging because the view range varies depending on the number of input views. Instead, in the same view range, I think varying the number of input views can verify whether more accurate geometry leads to better image quality.
> >
> > For example, how about comparing input views of 10, 30, 50, 70, 90, versus input views of 10, 20, 30, 40, 50, 60, 70, 80, 90? In this case, the latter one has more redundant gaussians as well as meaningful gaussians. So, it is interesting whether PTF handles more redundancy and semantic information than the former one, leading to better novel view quality.

---

> ### Author Response · Authors · 2024-08-09
>
> Thank you for your insightful question. To evaluate the effect of the number of input views when fixing the maximum view length, we conduct an experiment on ScanNet as follows: For each test scene, we first randomly pick the first frame (denoted as the *x-th* frame of the scene). Next, we set up three different view intervals (10, 20, 40) and fix the maximum view interval as 80, i.e. the chosen input views are [x, x+10, x+20, x+30, x+40, x+50, x+60, x+70, x+80] for view interval of 10, [x, x+20, x+40, x+60, x+80] for view interval of 20, and [x, x+40, x+80] for view interval of 40. For the target views, we select one target view between each interval: [x, x+10], [x+10, x+20], [x+20, x+30], ..., [x+70, x+80], and evaluate such target views for all the above settings to form a fair comparison. We evaluate the performance of our FreeSplat-*fv* as shown in the following Table:
>
>
>
> | Row  | View Interval |   PTF    | \# Gaussians (k) | Removed Gaussians (%) | PSNR&uarr; | SSIM&uarr; | LPIPS&darr; | $\delta<1.25$&uarr; |
> | :--: | :-----------: | :------: | ---------------- | --------------------- | :--------: | :--------: | :---------: | :-----------------: |
> | \#1  |     D=10      |          | 1769             | 0.0                   |   25.13    |   0.813    |    0.264    |        0.923        |
> | \#2  |     D=10      | &#10004; | 838              | 52.9                  | **25.70**  | **0.824**  |  **0.252**  |      **0.932**      |
> | \#3  |     D=20      |          | 983              | 0.0                   |   24.36    |   0.801    |    0.266    |        0.912        |
> | \#4  |     D=20      | &#10004; | 584              | 40.9                  |   24.79    |   0.815    |    0.255    |        0.920        |
> | \#5  |     D=40      |          | 590              | 0.0                   |   21.34    |   0.759    |    0.308    |        0.844        |
> | \#6  |     D=40      | &#10004; | 478              | 19.4                  |   21.53    |   0.766    |    0.300    |        0.847        |
>
>
>
> where D is the interval between nearby input views. The results indicate that when fixing the maximum view length, more input views can lead to better rendering quality and more accurate geometry. Furthermore, when encoding denser input views, our PTF module becomes more important in removing increasingly redundant Gaussians. This experiment is a very good example evaluating that more accurate geometry leads to better rendering quality, and the increasingly important role of our PTF module when given denser inputs. Note that when setting D=10 with PTF (row \#2), we give fewer Gaussians and +1.34dB PSNR comparing to D=20 without PTF (row \#3), which clearly demonstrates the necessity of PTF in benefiting from denser inputs while removing those redundant Gaussians. On the other hand, the significantly decreased results when setting D=40 also evaluate that inputting sparser input views similarly as RE10K or ACID for indoor scenes leads to unsatisfactory results. We hope the above experiment and analysis can address your question.

---

> > ### Comment · Reviewer_Hd37 · 2024-08-10
> >
> > Thank you for providing the results I was looking for. The experiments and clarifications provided in the rebuttal and the discussion period have addressed all of my concerns, and I appreciate the detailed answers to enhance my understanding of the paper. Thus, I will raise my score to weak accept.

---

### Official Review · Reviewer_emYN · 2024-07-16

**Soundness:** 2
**Presentation:** 3
**Contribution:** 2
**Rating:** 5
**Confidence:** 4

**Summary:**

1. Low-cost Cross-View Aggregation: This efficient methodology constructs adaptive cost volumes between proximate views and aggregates features utilizing a multi-scale structure. This approach enables the processing of extended input sequences and the incorporation of more stringent geometric constraints.
2. Pixel-wise Triplet Fusion (PTF): This module aligns and merges local Gaussian triplets into global representations, thereby mitigating redundancy in overlapping regions and consolidating features observed across multiple viewpoints.
3. Free-View Training (FVT): This novel training strategy decouples the model's performance from a predetermined number of input views, thus enabling robust view synthesis across an expanded range of viewpoints.
4. The authors demonstrate that FreeSplat surpasses existing methodologies in both novel view synthesis quality and depth map accuracy on the ScanNet and Replica datasets. The proposed method exhibits enhanced performance in view interpolation and extrapolation tasks, particularly for extended input sequences. Moreover, FreeSplat reduces the quantity of redundant Gaussians and offers more computationally efficient inference compared to preceding approaches.

**Strengths:**

1. The paper is well-structured, offering lucid explanations of the technical approach. It includes detailed architectural diagrams and pseudocode to enhance reproducibility. Comprehensive ablation studies are presented to isolate the impact of different components. The overall writing is coherent and accessible.
2. This research enables free-viewpoint rendering and 3D reconstruction from arbitrary numbers of input views, broadening the applicability of 3D Gaussian splatting techniques. It enhances efficiency and minimizes redundancy in 3D Gaussian representations for extended sequences.

**Weaknesses:**

1. Experimental Scope and Comparability: PixelSplat presents experimental results on the Re10k and ACID datasets. For a comprehensive and equitable comparison, it would be beneficial for the proposed method to also report results on these two datasets. This would facilitate a direct comparison with pixelSplat across multiple benchmarks, thereby strengthening the validity of the authors' claims. The absence of experimental results on Re10k and ACID raises questions about the proposed method's performance and generalizability across diverse datasets. I am wondering why the experimental results on Re10k  and ACID are absent.
2. Analysis of Failure Cases and Limitations: The paper would be significantly enhanced by a more thorough examination of failure cases and limitations of the proposed approach. While the appendix briefly touches on this aspect, a more in-depth analysis would provide valuable insights into the method's robustness and potential areas for improvement. Such an analysis could include specific examples of scenarios where the method underperforms, a discussion of the underlying causes for these failures, and potential strategies for addressing these limitations in future work. This level of critical self-evaluation would not only increase the paper's scientific rigor but also provide a more balanced perspective on the method's capabilities and constraints.

**Questions:**

Please see the weaknesses.

**Limitations:**

The authors discuss some limitations of their work:
1. High GPU memory requirements (>40GB) for extremely long input sequences (>50 images).

2. A gap in 3D reconstruction accuracy compared to state-of-the-art methods that use 3D supervision or RGB-D inputs.

---

> ### Author Rebuttal · Authors · 2024-08-06
>
> ## To Reviewer emYN (#R2):
>
> 1. **Experiments on Re10k and ACID:** To further evaluate our model's generalization ability across diverse domains, we train our model on RE10K using 2-View setting and 5-View setting respectively. The results are shown in our ***rebuttal pdf Table 2, 3 and Figure 3***. Note that for the 5-View setting inference, we sample input views with random intervals between 25 and 45 due to the limited sequence lengths in RE10K and ACID. In the 2-View setting, we perform better than pixelSplat [1] and on par as MVSplat [2] on both datasets. In the 5-View setting, we outperform both baselines by a clear margin. We analyze the main causes of the above results as follows:
>
>    In the 2-view comparison experiments with the baselines, the image interval between the given stereo images were set to be large. On average, the interval between image stereo is 66 in RE10K and 74 in ACID, which is much larger than our indoor datasets setting (20 for ScanNet and 10 for Replica). Such large interval can result in **minimum view overlap** between the image stereo (e.g. as shown in our ***rebuttal pdf Figure 4(b)***), which means that our cost volume can be **much sparser** and multi-view information aggregation is weakened. In contrast, MVSplat uses a cross-view attention that aggregates multi-view features through a sliding window which does not leverage camera poses. pixelSplat uses a heavy 2D backbone that can potentially become stronger monocular depth estimator. In our 5-view setting, we outperform both baselines by clear margins. This is partially due to the smaller image interval and larger view overlap between nearby views. As a result, our cost volume can effectively aggregate multi-view information, and our PTF module can perform point-level fusion and remove those redundant 3D Gaussians.
>
>    Therefore, our model is not specifically designed for highly sparse view inputs, but it is designed as a low-cost model that can easily take in much longer sequences of higher-resolution inputs, that is suitable for indoor scene reconstruction (we also offer a quantitative comparison on computation cost in our ***rebuttal Table 1*** to emphasize our strengths). Comparing to RE10K and ACID, real-world indoor scene sequences usually contain more complicated camera rotations and translations, which results in the requirement of more dense observations to reconstruct the 3D scenes with high completeness and accurate geometry. Consequently, our model is targeting the fast indoor scene reconstruction with keyframe inputs, which contain long sequences of high-resolution images, while existing works struggle to extend to such setting as evaluated in our main paper.
>
>    We really appreciate your question, which also helped us delve into the comparison on a larger experimental scope and analyze the reasons behind the results. We hope that our analysis can help demonstrate our contributions more clearly. We will add the corresponding results and the above illustrations in our final version paper to improve the completeness of our paper and highlight the contributions of our paper comparing to existing works.
>
> 2. **Failure cases and potential future works:** As shown in our ***rebuttal pdf Figure 3***, we visualize the whole scene reconstruction results highlighting (a) the errorenously estimated depth for the specular/texture-less regions, and (b) Difficulty of accurate depth estimation when the given input stereo has extremely large interval. Such errors are mainly due to the following aspects:
>
>    **(1) Lack of depth regularizations.** The nature of our color-supervised depth estimation methods makes it difficult to accurately estimate depth for such regions. One potential solution is to leverage depth supervision / priors, e.g. regularize the depth estimation results using GT depth / sparse depth from COLMAP / monocular depth estimation methods. Future works can also explore the addition of geometric constraints on Gaussians localisation (eg. depth smoothness regularization), or multi-view consistency regularizations to enhance the 3D Gaussian localization.
>
>    **(2) Straightforward 3D Gaussian fusion method.** Although we have proposed the PTF module to reduce redundant 3D Gaussians and improve the depth estimation performance, it is still not enough to reach satisfactory multi-view fusion results. Future works can learn from TSDF fusion methods [3] and enhance the unprojection of 3D Gaussians, e.g. given the initial 3D Gaussians from the first frame, search for its projections on all the remaining frames, and unproject them together.
>
>    **(3) Ineffectivity of MVS encoder when faced minimum view overlap between inputs:** When the given inputs only have minimum view overlap, our MVS can hardly find correspondences on the reference view, which results in highly sparse cost volume and ineffectivity of multi-view feature aggregation. It would be beneficial to leverage a cross-view attention (e.g. as in MVSplat) which can work despite minimum view overlap. Although our transformer-free backbone may underperform in such extreme cases, it was specifically designed for relatively dense input of long sequences. In the main paper, we have evaluated that our low-cost backbone is essential for accurate indoor scene reconstruction.
>
>    We will add the above illustrations in our final version to delve deeper into the analysis of failure cases and discuss potential future works that can be explored upon our method.
>
> [1] Charatan, David, et al. "pixelsplat: 3d gaussian splats from image pairs for scalable generalizable 3d reconstruction." CVPR. 2024.
>
> [2] Chen, Yuedong, et al. "Mvsplat: Efficient 3d gaussian splatting from sparse multi-view images." arXiv preprint arXiv:2403.14627 (2024)."
>
> [3] Choe, Jaesung, et al. "Volumefusion: Deep depth fusion for 3d scene reconstruction." *CVPR*. 2021.

---

> > ### Comment · Reviewer_emYN · 2024-08-13
> > **I change my decision to boarderline accept**
> >
> > Your response addressed most of my concerns, and I have decided to raise my score after reading your reply and your discussion with other reviewers.

---

### Official Review · Reviewer_gTxT · 2024-07-17

**Soundness:** 3
**Presentation:** 3
**Contribution:** 3
**Rating:** 6
**Confidence:** 3

**Summary:**

This paper proposes FreeSplat to reconstruct geometrically consistent 3D scenes from long sequence inputs. To this end, the paper presents Low-cost Cross-View Aggregation for feature matching and Pixel-wise Triplet Fusion for Gaussian triplets fusion. The outstanding results of long sequence 3DGS generalization shows the superiority beyond baseline methods.

**Strengths:**

1. The paper is well-written and the technical contributions are clearly clarified and easy to understand.
2. The experimental results over baselines are attractive and convincing, which demonstrates the effectiveness of the proposed method.
3. The proposed pixel-wise triplet fusion is reasonable and novel.

**Weaknesses:**

1. The low-cost cross-view aggregation module is not novel enough. It seems that the three submodules (2D feature extraction, cost volume formulation, multi-scale feature aggregation) are all borrowed from existing methods and the low-cost cross-view aggregation is a simple stack of these submodules. I suggest the author to further clarify the differences between the proposed module and the existing MVS methods.

**Questions:**

1. What are the differences between Freeplat-spec and Freesplat-fv?
2. The author provided 3-view version results in the long sequence reconstruction experiment in Table 2. I’m doubting why do you provide the 10-view version of the baselines? Just because their original paper did not provide the 10-view version, or because they will fail to reconstruct a scene under 10-view inputs, or other reasons? Further clarification will help to understand this experiment.
3. The author claims that the proposed pixel-wise triplet fusion module helps to remove redundant Gaussians in the overlapping regions and the experimental results of final Gaussian number demonstrates this. However, it is somewhat not intuitive. It would be better to provide a visualization to show how the Gaussian ellipsoids are reduced in the overlapping regions.

**Limitations:**

The author discussed the limitations in the appendix about the memory consumption and the limited performance of generalizable Gaussians, which I think is valuable. It remains a significant problem for further study.

---

> ### Author Rebuttal · Authors · 2024-08-06
>
> ## To Reviewer gTxT (#R1):
>
> 1. **Differences from existing MVS methods:** Compared to traditional MVS methods [1,2], the main difference of our backbone lies in the unsupervised scheme of depth estimation supervised purely by color images, while reaching comparable depth estimation accuracy. Due to the unsupervised nature of our method, the pixel-aligned depth values are predicted through weighted summing along the candidate depth planes instead of predicting absolute depth values, in order to bound the depth values within a reasonable range. Compared to existing MVS-based NeRF methods [3, 4], our backbone design avoids expensive 3D CNN / ray transformer, and our explicit 3DGS reconstruction can greatly benefit from real-time renderings. Compared to MVS-based 3DGS methods [5, 6], our main contributions lie on designing a transformer-free 2D backbone to encode extended input sequences of high-resolution images, and adaptively formulate cost volumes within nearby views based on pose affinities. More importantly, we are the first to explore explicit long sequence reconstruction based on generalizable 3DGS. Both pixelSplat and MVSplat struggle to extend to long sequences of high-resolution inputs due to their patch-wise 2D Transformers / cross-view Transformers, which consume quadratically increasing GPU memory with image sequence length and resolution. On the other hand, our low-cost 2D backbone is designed to easily encode such indoor scene sequences. Therefore, although our 2D backbone bears high-level design similarity to existing MVS methods, the differences in our design are specifically targeting long sequence reconstruction with precise unsupervised depth estimation which we demonstrated to be significant for our goal.
> 2. **Differences between FreeSplat-spec and FreeSplat-fv:** FreeSplat-spec is trained with fixed number of input views (eg. 2, 3), and FreeSplat-fv is trained using our free-view training strategy using *2~8* input views, in order to extend our method to arbitrary length of inputs. We will include the above explanation in the final version.
> 3. **Why providing 3-view version of baselines in Table 2:** The main reason is the enormous GPU consumption of pixelSplat and MVSplat when inputting more views. As shown in our ***rebuttal pdf Table 1***, pixelSplat and MVSplat already require **30~50 GB** GPU when training with 3 input views and batch size of 1 due to their heavy patch-based 2D Transformers and cross-view attention. Their GPU requirements quadratically increase with respect to the inputs resolution and length, thus making training their 10-views version nearly infeasible. In contrast, our model can train on 8 input views while requiring a smaller GPU memory compared to 3 views version of pixelsplat and MVSplat, supporting training and inference on long sequences of high-resolution inputs. Therefore, in our main paper Table 2, we report pixelSplat and MVSplat's 10-views inference results using their 3-views trained models due to the infeasibility of training their models using more views. To form a fair comparison, we also compare our 3-view version on 10-view inference, where we consistently outperform the 3-views baselines. On the other hand, the further improvements brought by our FreeSplat-fv over our 3-view version are creditable to our low-cost backbone design.
> 4. **Visualization of PTF for removing redundant 3D Gaussians:** To illustrate our proposed PTF more intuitively, we draw an illustration figure in our ***rebuttal pdf Figure 1***, and visualize the fusion process as shown in our ***rebuttal pdf Figure 2***. Our proposed PTF module can greatly remove the redundant 3D Gaussians that lie very closely to the existing ones, such that we avoid redundant unprojection of 3D Guassians to the regions that are observed multiple times. PTF can also fuse the redundant 3D Gaussians latent features to aggregate multi-view observations at point-level, alleviating artifacts caused by lightning conditions, etc. Furthermore, as shown in our ***rebuttal pdf Table 1***, applying PTF module can increase the rendering speed from **39 to 72 FPS** during whole scene reconstruction, which becomes more essential when the 3D scene is larger with more input views. Since generalizable 3D Gaussians methods normally unproject pixel-wise 3D Gaussians for each input view that can easily result in redundancy, removing the redundant ones become more important to achieve real-time rendering.
>
> [1] Yao, Yao, et al. "Mvsnet: Depth inference for unstructured multi-view stereo." Proceedings of the European conference on computer vision (ECCV). 2018.
>
> [2] Sayed, Mohamed, et al. "Simplerecon: 3d reconstruction without 3d convolutions." European Conference on Computer Vision. Cham: Springer Nature Switzerland, 2022.
>
> [3] Chen, Anpei, et al. "Mvsnerf: Fast generalizable radiance field reconstruction from multi-view stereo." Proceedings of the IEEE/CVF international conference on computer vision. 2021.
>
> [4] Wang, Qianqian, et al. "Ibrnet: Learning multi-view image-based rendering." Proceedings of the IEEE/CVF conference on computer vision and pattern recognition. 2021.
>
> [5] Charatan, David, et al. "pixelsplat: 3d gaussian splats from image pairs for scalable generalizable 3d reconstruction." Proceedings of the IEEE/CVF Conference on Computer Vision and Pattern Recognition. 2024.
>
> [6] Chen, Yuedong, et al. "Mvsplat: Efficient 3d gaussian splatting from sparse multi-view images." arXiv preprint arXiv:2403.14627 (2024)."

---

> > ### Comment · Reviewer_gTxT · 2024-08-09
> > **About Figure 2**
> >
> > Thanks for your rebuttal! Sorry for the typo in my review. "I’m doubting why do you provide the 10-view version of the baselines?" should be "I’m doubting why don't you provide the 10-view version of the baselines?", but your rebuttal has solved my question. I am still unclear about Figure 2 in the rebuttal PDF. Do you mean that all of the Gaussians in the blue mask are fused and removed? Why are there still some excluded Gaussians in the second image? Is it because they are retained after the fusion process? Could you give some more clarification on this?

---

> > > ### Author Response · Authors · 2024-08-09
> > >
> > > We are glad to know that your previous question has been solved. Regarding PTF fusion visualization, the blue mask denotes the removed Gaussians after the PTF process. We can see that most of them lie on the sofa and the floor since our depth estimation for those regions is more accurate and the PTF can effectively fuse those Gaussians that lie very closely to the existing ones. There are also some uncovered regions in the second image, which means that those regions do not contain the removed Gaussians. This is because the depth estimation for those regions are not sufficiently accurate. For example, as shown in our rebuttal pdf Figure 4(a), the estimated depth for the red wall (denoted within the blue box) is not sufficiently accurate. This is due to the difficulty of MVS-based depth estimation on the texture-less appearance of the wall. Consequently, the wall region is largely uncovered in our visualized fusion process (Figure 2 in the rebuttal pdf) since our PTF uses a threshold on the difference between the local Gaussians and the global Gaussians to determine the fusion. To this end, it is also possible to set higher thresholds in PTF to reduce more Gaussians, or future works can explore adding more depth supervision (e.g. GT depth / COLMAP coarse depth) or multi-view consistency to further regularize the Gaussian localization. We also provide brief illustrations regarding this part in our reply to Reviewer emYN (#R2) point 2, where we analyze the failure cases and potential future works. We hope that the above illustration can help solve your question, and we are willing to provide clarifications on any further questions.

---

> > > > ### Comment · Reviewer_gTxT · 2024-08-09
> > > >
> > > > Thanks for the clarification. I now understand it. The rebuttal has addressed all of my concerns.  I think it is a valuable paper to be accepted by NeurIPS, and as such, I am willing to raise my rating from borderline accept to weak accept.

---

> > > > > ### Comment · Area_Chair_FDxA · 2024-08-09
> > > > >
> > > > > Thanks for reviewer gTxT's comments. Do other reviewers have some comments?
> > > > >
> > > > > Best,
> > > > >
> > > > > AC

---

### Author Rebuttal · Authors · 2024-08-06

## To all Reviewers:

We first thank all reviewers for your valuable time and inspiring comments. As summarized by our reviewers, our proposed method is "reasonable and novel" (\#R1) and "interesting" (\#R3), and our experimental results are "attractive and convincing" (\#R1), "exhibits enhanced performance" (\#R2), "advance in long sequence novel view synthesis" (\#R4).

Regarding the raised questions, we have conducted extensive experiments as shown in our rebuttal pdf, and we provide corresponding introductions as follows:

1. **Comparison of computational cost and whole scene reconstruction:** As shown in our ***rebuttal pdf Table 1***, we compare the required GPU memory for training and testing, the encoding time, rendering FPS, and PSNR for whole scene reconstruction. pixelSplat-3views and MVSplat-3views already consume *30~50 GB* GPU memory for training due to their quadratically increasing GPU memory requirement w.r.t image resolution / sequence length. Therefore, it becomes infeasible to extend their methods to higher resolution inputs or longer sequence training. In comparison, our low-cost framework design enable us to effectively train on long sequence inputs while requiring lesser GPU memory compared to the 3 views version of existing methods. Furthermore, our proposed PTF module can effectively reduce redundant 3D Gaussians, improving rendering speed from 39 to 72 FPS . This becomes increasingly important when reconstructing larger scenes since generalizable 3DGS normally perform pixel-wise unprojection, which can easily result in redundancy in the overlapping regions (as shown in our ***rebuttal pdf Figure 2***).

2. **Experiments on RE10K and ACID:** To further evaluate our model's generalization ability across diverse domains, we train our model on RE10K using 2-View setting and 5-View setting respectively. The results are shown in our ***rebuttal pdf Table 2, 3 and Figure 3***. Note that for the 5-View setting inference, we sample input views with random intervals between 25 and 45 due to the limited sequence lengths in RE10K and ACID. In the 2-View setting, we perform better than pixelSplat [1] and on par as MVSplat [2] on both datasets. In the 5-View setting, we outperform both baselines by a clear margin. We analyze the main causes of the above results as follows:

   In the 2-view comparison experiments with the baselines, the image interval between the given stereo images were set to be large. On average, the interval between image stereo is 66 in RE10K and 74 in ACID, which is much larger than our indoor datasets setting (20 for ScanNet and 10 for Replica). Such large interval can result in **minimum view overlap** between the image stereo (e.g. as shown in our ***rebuttal pdf Figure 4(b)***), which means that our cost volume can be **much sparser** and multi-view information aggregation is weakened. In contrast, MVSplat uses a cross-view attention that aggregates multi-view features through a sliding window which does not leverage camera poses. pixelSplat uses a heavy 2D backbone that can potentially become stronger monocular depth estimator. In our 5-view setting, we outperform both baselines by clear margins. This is partially due to the smaller image interval and larger view overlap between nearby views. As a result, our cost volume can effectively aggregate multi-view information, and our PTF module can perform point-level fusion and remove those redundant 3D Gaussians.

   Therefore, our model is not specifically designed for highly sparse view inputs, but it is designed as a low-cost model that can easily take in much longer sequences of higher-resolution inputs, that is suitable for indoor scene reconstruction (we also offer a quantitative comparison on computation cost in our ***rebuttal Table 1*** to emphasize our strengths). Comparing to RE10K and ACID, real-world indoor scene sequences usually contain more complicated camera rotations and translations, which results in the requirement of more dense observations to reconstruct the 3D scenes with high completeness and accurate geometry. Consequently, our model is targeting the fast indoor scene reconstruction with keyframe inputs, which contain long sequences of high-resolution images, while existing works struggle to extend to such setting as evaluated in our main paper.

3. **Comparison with SurfelNeRF:** We further compare with SurfelNeRF as shown in our ***rebuttal pdf Table 3 and Figure 4***. We evaluate on the same novel views as theirs, sampling input views along their input sequences with an interval of 20 between nearby views. Note that the number of input views changes when the input length changes, while our FreeSplat-*fv* can seamlessly conduct inference with arbitrary numbers of inputs. Our method performs significantly better than SurfelNeRF in both rendering quality and efficiency. Our end-to-end framework jointly learns depths and 3DGS using an MVS-based backbone, while SurfelNeRF relies on depths and does not aggregate multi-view features to assist their surfel feature prediction.

[1] Charatan, David, et al. "pixelsplat: 3d gaussian splats from image pairs for scalable generalizable 3d reconstruction." Proceedings of the IEEE/CVF Conference on Computer Vision and Pattern Recognition. 2024.

[2] Chen, Yuedong, et al. "Mvsplat: Efficient 3d gaussian splatting from sparse multi-view images." arXiv preprint arXiv:2403.14627 (2024)."

---

### Comment · Area_Chair_FDxA · 2024-08-07

Hi reviewers,

Thanks a bunch for all your hard work as reviewers for NeurIPS 2024.

The discussion period between reviewers and authors has started. Make sure to check out the authors' responses and ask any questions you have to help clarify things by 8.13.

Best,
AC

---

> ### Comment · Area_Chair_FDxA · 2024-08-12
>
> Dear reviewers,
>
> As the reviewer-author discussion period is about to end by 8.13, please take a look at other reviewers' reviews and authors' rebuttal at your earliest convenience. It would be great if you could ask authors for more clarification or explanation if some of your concerns are not addressed by the rebuttal.
>
> Thanks,
>
> AC

---

### Author Response · Authors · 2024-08-12

## To All Reviewers:

We sincerely thank all reviewers for their valuable comments and feedback, and we believe that we have provided detailed experiments and illustrations to thoroughly address all concerns in our responses. We encourage reviewers to check all of our responses and to see whether to ask for any additional clarifications before the discussion phase concludes. Below is a brief summary of our additional responses during the discussion period:

1. **Explanations of our PTF visualization**. In our replies to **Reviewer gTxT (\#R1)**, we provided detailed explanations regarding the PTF visualization included in our rebuttal pdf.
2. **Effect of the number of input views**. In our replies to **Reviewer Hd37 (\#R3)**, we discussed about the effect of the number of input views when fixing the maximum view interval. We found that when fixing the maximum view interval and increasing the number of input views, both the geometry accuracy and rendering quality improve.
3. **Comparison with 3DGS-based SLAM method**. In our replies to **Reviewer ZuRj (\#R4)**, we compared our method with 3DGS-based SLAM method MonoGS [1]. We found that our method is capable of performing whole scene reconstruction more effectively and much more efficiently, which further demonstrates the value of our work.

Overall, we believe our work can bring valuable impacts to the community, and hope that we can get a chance to share our work with everyone at NeurIPS 2024.




[1] Matsuki, Hidenobu, et al. "Gaussian splatting slam." *Proceedings of the IEEE/CVF Conference on Computer Vision and Pattern Recognition*. 2024.

---

### Decision · Program_Chairs · 2024-09-25

**Decision:**

Accept (poster)

**Comment:**

The submission got four positive recommendations. The reviewers were concerned about the experiments, evaluations, limitations, missing details, and more analysis on the methods. The authors did a good job in the rebuttal and had good discussions with reviewers during the reviewer and author discussion period, both of which are helpful to address most reviewers’ concerns. The reviewers reached a consensus of acceptance before the AC and reviewers discussion period. The AC read through the paper, review, rebuttal, and the discussion. The AC supports the reviewers’ decision, and makes a decision to accept this submission. This decision was approved by the SAC.